# Position: Evaluation of ECG Representations Must Be Fixed

Zachary Berger [1 2 *]   Daniel Prakah-Asante [1 2 *]   John Guttag [1]   Collin M. Stultz [1 2]

## Abstract

This position paper argues that current benchmarking practice in 12-lead ECG representation learning must be fixed to ensure progress is reliable and aligned with clinically meaningful objectives. The field has largely converged on three public multi-label benchmarks (PTB-XL, CPSC2018, CSN) dominated by arrhythmia and waveform-morphology labels, even though the ECG is known to encode substantially broader clinical information. We argue that downstream evaluation should expand to include an assessment of structural heart disease and patient-level forecasting, in addition to other evolving ECG-related endpoints, as relevant clinical targets. Next, we outline evaluation best practices for multi-label, imbalanced settings, and show that when they are applied, the literature's current conclusion about which representations perform best is altered. Furthermore, we demonstrate the surprising result that a randomly initialized encoder with linear evaluation matches state-of-the-art pre-training on many tasks. This motivates the use of a random encoder as a reasonable baseline model. We substantiate our observations with an empirical evaluation of five representative ECG pre-training approaches across six evaluation settings: the three standard benchmarks, a structural disease dataset, hemodynamic inference, and patient forecasting. Code is available at `https://github.com/zackeberger/ecg-fix`.

## 1. Introduction

Representation learning aims to produce features that are useful across many downstream applications. This reduces reliance on large labeled datasets to learn task-specific mod-els (Bengio et al., 2013). However, there is a tension between learning features that are broadly useful versus those that excel for particular tasks (Bommasani et al., 2022). This tension is pronounced in medicine, where labeled data can be sparse and clinical use-cases varied (Esteva et al., 2019).

In medicine, what constitutes a meaningful endpoint task depends on the modality. For example, the clinical targets of chest X-ray differ from those of electroencephalography. As a result, benchmarking practice in medicine must be discussed on a per-modality basis. These choices shape which representations appear effective and steer subsequent methodological development (Lipton & Steinhardt, 2019; Pineau et al., 2021).

Here, we focus on the 12-lead electrocardiogram (ECG), an inexpensive and commonly used diagnostic tool that non-invasively records the heart's electrical activity (Noble et al., 1990). ECG representation learning is an active area of research and has recently received attention in major ML venues (e.g., ICML, ICLR, NeurIPS, AAAI) (Liu et al., 2024b; Hung et al., 2024; Wang et al., 2025; Na et al., 2024; Jin et al., 2025; Chen et al., 2025; Lan et al., 2022). As the literature grows, we argue it is timely to re-examine the field's current benchmarking practice. A rigorous and uniform benchmarking strategy is essential to ensure reliable and reproducible results, and most importantly that the learned representations align with clinically meaningful objectives. ECG benchmarking is associated with a number of challenges, including the use of large multi-label and severely imbalanced datasets – issues common across many applied machine learning (ML) settings, and especially medicine (Zhang & Zhou, 2014; He & Garcia, 2009).

This position paper examines how current task selection and reporting practice shape conclusions about ECG representation quality. We then evaluate these choices empirically across five representative pre-training methods and six evaluation settings.

In Section 2, we observe that three datasets have become standard for evaluating 12-lead ECG representations: PTB-XL (Wagner et al., 2020), CPSC2018 (Liu et al., 2018), and CSN (Zheng et al., 2020). Each is a multi-label suite focused primarily on binary arrhythmia outcomes and waveform morphology classification. However, the ECG has increasingly been recognized to contain information perti-

*Equal contribution [1]Massachusetts Institute of Technology, Cambridge, MA, USA [2]Massachusetts General Hospital, Boston, MA, USA. Correspondence to: Zachary Berger <zberger@mit.edu>.

*Proceedings of the 43rd International Conference on Machine Learning*, Seoul, South Korea. PMLR 306, 2026. Copyright 2026 by the author(s).

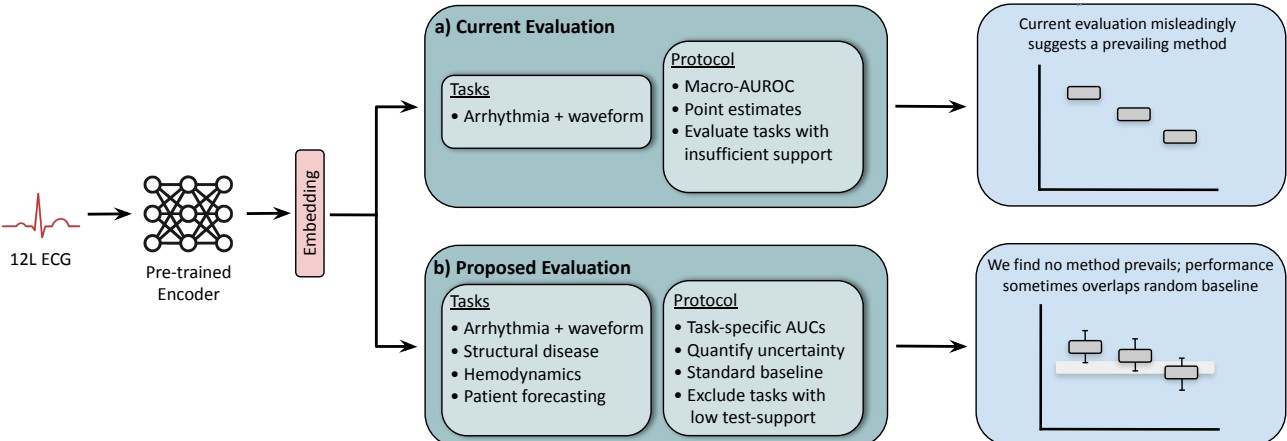

*Figure 1.* Overview of the evaluation pipeline for 12-lead ECG representations. (a) Current practice focuses on arrhythmia/waveform tasks and macro-AUROC point estimates, which can produce misleading method rankings. (b) We propose a broader set of clinically relevant tasks and evaluation best-practices that more reliably assess methods. We find that no method consistently prevails and for many tasks, many methods overlap with the baseline of a randomly initialized encoder.

nent to a wider variety of outcomes, e.g., structural disease (Poterucha et al., 2025), hemodynamic state (Schlesinger et al., 2022), and patient forecasting (Khurshid et al., 2022; Bergamaschi et al., 2025). In Section 3, we propose additional tasks that would be a welcome addition in the evaluation pipeline. This is particularly relevant as the field moves towards more complex tasks, where ECGs are combined with other modalities to predict clinical outcomes that are not deterministic functions of the ECG alone.

To manage multi-label evaluation across many tasks, the field has largely converged on summarizing performance with macro-AUROC, which aggregates per-label AUROCs through an unweighted mean (Zhang & Zhou, 2014). This convention enables straightforward comparison across methods, but obscures clinically meaningful behavior. Clinicians ultimately deploy models for specific purposes, yet macro-AUROC masks performance on individual endpoints by collapsing them into a single number. This issue is compounded by severe label imbalance; many ECG labels have few positive examples, yielding noisy task-level estimates. However, uncertainty is seldom reported alongside headline metrics. Moreover, AUROC alone can misrepresent performance on imbalanced labels, where alternative metrics may better reflect clinical utility (Davis & Goadrich, 2006; Saito & Rehmsmeier, 2015). In Section 4, we suggest a set of reporting and evaluation best practices and show that many prior studies do not adhere to them.

In Section 5, we show that applying these practices can change method rankings on standard benchmarks and alter conclusions of the current literature about which methods perform best. We also show that a randomly initialized encoder with linear evaluation matches the performance of state-of-the-art ECG pre-training methods on many tasks.

Our position is visualized in Figure 1. We argue that going forward, evaluation of representations of ECG should

- Cover a wider variety of tasks than is currently typical. For example, they should include prediction of patient outcomes or estimates of structural disease rather than just arrhythmia and waveform classification.

- Report clinically relevant task-specific findings rather than focus, as most papers do, on aggregate performance. For example, they should report on AUROC, precision, and recall for individual tasks rather than just macro-AUROC over classes of tasks.

- Carefully characterize uncertainty, which can be quite high for tasks with a small number of positive examples, which is common in ECG datasets.

- Compare the utility of learned representations to that of a simple baseline: a randomly initialized encoder.

## 2. Related Work

**Benchmarking in ML.** Progress in empirical ML has long been driven by benchmarks (e.g., Geiger et al., 2012; Lin et al., 2014; Russakovsky et al., 2015; Wang et al., 2018). However, evaluation practices often lack rigor and standardization (Lipton & Steinhardt, 2019; Liao et al., 2021; Herrmann et al., 2024). This has motivated work that critically examines and improves benchmark design and reporting (e.g., Gebru et al., 2021; Pineau et al., 2021; Vendrow et al., 2025). Many lessons have emerged from domain-specific settings, for instance, in recommendation systems (Ferrari Dacrema et al., 2019), neural network pruning (Blalock et al., 2020), anomaly detection (Liu & Paparrizos, 2024), and graph learning (Bechler-Speicher et al., 2025).

**ECG Benchmarking.** ECGs are routinely collected in clinical care, so large labeled corpora exist in many health systems. Yet, these data are rarely shared because of patient privacy constraints and institutional requirements. As a result, despite the volume of ECGs that exist, there are few open-source datasets.

In response, the community has largely repurposed the available public datasets for downstream benchmarking of ECG representations. Evaluations typically center around three datasets, CPSC2018 (Liu et al., 2018), PTB-XL (Wagner et al., 2020; 2022), and CSN (Zheng et al., 2020; 2022), which contain on the order of tens of thousands of recordings. All three focus narrowly on arrhythmia and waveform abnormality labels. Several other datasets with similar labels are occasionally used for testing, or aggregated together for training (e.g., Perez Alday et al., 2022; Liu et al., 2022; Ribeiro et al., 2020).

MIMIC-IV (Gow et al., 2023; Goldberger et al., 2000) and CODE-15 (Ribeiro et al., 2020) are commonly used for pre-training, since they are large open-source datasets. MIMIC-IV is of particular importance because its ECGs are linked to electronic health record data. This has enabled development of learning algorithms that incorporate clinical context through multi-modal supervision. Such approaches have recently garnered state-of-the-art performance (Liu et al., 2024b; Hung et al., 2024). Many studies pre-train on private institutional data (e.g., Diamant et al., 2022) or semi-restricted resources (e.g., Sudlow et al., 2015; Littlejohns et al., 2020; Koscova et al., 2024). While these datasets can be valuable for scaling up training data, their restricted access limits reproducibility.

There are few open-access datasets that expand beyond arrhythmia and waveform abnormality labels. A notable recent release is EchoNext, which links ECGs to echocardiography-derived ground truth to study structural heart disease (Poterucha et al., 2025).

Several preprints contemporaneous to this work have taken initial steps toward improving benchmarking for ECG representation learning (Lunelli et al., 2025; Al-Masud et al., 2025; Wan et al., 2025). These efforts each aim to consolidate evaluation practices in an open-source framework. However, they do not implement all of the protocol recommendations we discuss in Section 4. They also do not capture the breadth of clinically grounded tasks we argue the field should work toward in Section 3.

**ECG Representation Learning.** Self-supervised pre-training is a standard paradigm, driven by successes in computer vision (Chen et al., 2020) and natural language processing (Devlin et al., 2019). Early ECG representation learning (Kiyasseh et al., 2021; Mehari & Strodthoff, 2022) adapted popular vision frameworks such as SimCLR (Chen et al., 2020) and BYOL (Grill et al., 2020). These are now widely used as baselines in the ECG literature.

Many specialized methods have introduced inductive biases specific to 12-lead ECGs. These often take inspiration from contrastive learning and reconstruction-based learning. Contrastive methods learn representations by bringing together related samples while maximizing the distance between unrelated samples. These positive and negative pairs can be generated through augmentation (Chen et al., 2020; Grill et al., 2020; Chen & He, 2020; Chen et al., 2021) or by relying on information such as patient identity (Diamant et al., 2022). CLOCS (Kiyasseh et al., 2021) is a popular ECG-specific approach that builds pairs directly from the temporal and lead-structure of the signal. Reconstruction-based methods optimize representations by compressing then reconstructing examples directly, often by masking then filling in part of the signal (He et al., 2021; Na et al., 2024; Zhang et al., 2023a;b). HeartLang (Jin et al., 2025) is a recent example that leverages an ECG-specific tokenizer and trains using both reconstruction and masked-token prediction.

In parallel, recent work uses multi-modal supervision, commonly pairing ECGs with clinical text from the electronic health record (Lalam et al., 2023; Yu et al., 2024). MERL (Liu et al., 2024b), D-BETA (Hung et al., 2024), and KED (Tian et al., 2024) are recent examples that have claimed state-of-the-art performance following this approach. MERL aligns ECG embeddings with representations of their paired text reports using a contrastive objective. D-BETA extends this work by regularizing the learning process with reconstruction loss on the text and ECG. KED enriches the text supervision with clinical knowledge generated by a large language model.

In this paper, we characterize the state of benchmarking for ECG representation learning by surveying work published at major ML conferences since 2019. We also include approaches cited by those publications. When making broad claims about the field, we refer to this *survey set* of 28 methods; details are provided in Appendix A. We note that many more pre-trained ECG models have been proposed, some of which are covered in the review of (Han et al., 2025).

For our empirical study, we focus on CLOCS, KED, HeartLang, MERL, and D-BETA as exemplar methods. They span several prominent paradigms in recent ECG representation learning, including contrastive learning, reconstruction, and multi-modal ECG-text supervision. CLOCS and MERL are widely cited, while HeartLang, KED, and D-BETA are recent works reporting state-of-the-art performance. All five are supported by released weights or pre-training code.

## 3. Extending Current Benchmarks

Historically, ECG interpretation has centered on rhythm and waveform abnormalities (Rivera-Ruiz et al., 2008; Fisch, 2000). As a result, downstream evaluation of ECG representations has largely focused on such tasks. In our survey of 28 ECG representation learning papers, 23 report results on PTB-XL, 15 on CPSC2018, 11 on CSN or its constituent datasets (Chapman-Shaoxing and Ningbo), with 25 evaluating on at least one of the three (see Table 8).

The ground-truth labels for arrhythmia and waveform abnormality tasks are obtained from inspection of an ECG by an expert. Thus, all information needed to assign the label is explicitly contained in the signal. Yet, it has recently been shown that machine-learned models can be applied to the ECG to infer clinically relevant endpoints that are not readily visible in the signal (Friedman et al., 2025). Downstream evaluation of ECG representations should expand to better reflect this clinical scope.

We propose the following families of downstream tasks be tested in evaluations. This categorization is motivated by the distinction between tasks whose labels are assigned directly from the ECG trace and tasks whose ground truth is obtained from paired measurements, such as imaging and catheterization, or from future clinical outcomes.

1. **Arrhythmia and Waveform Abnormalities** include tasks derived from expert interpretation of the ECG trace. Many public datasets, e.g., PTB-XL, CPSC2018, and CSN (Wagner et al., 2020; Liu et al., 2018; Zheng et al., 2020) already include relevant labels.

2. **Structural Disease** includes tasks that probe cardiac morphology or function, such as systolic function and valvular disease. Labels for these are often derived from contemporaneous imaging including echocardiography and cardiac MRI. EchoNext is an open-source dataset of paired ECG and echocardiogram findings containing relevant labels (Poterucha et al., 2025).

3. **Hemodynamic State** targets inference of cardiac filling pressures, e.g., mean pulmonary capillary wedge pressure (mPCWP) and flows (Schlesinger et al., 2022). Ground truth for these labels often comes from right heart catheterization. Much of the literature on these tasks uses proprietary data. Publicly, MIMIC-IV contains some paired bedside-monitor ECG and invasive blood pressure signals (Moody et al., 2022).

In addition to task family, downstream targets can be separated into *diagnosis* and *patient forecasting*. *Diagnosis* involves estimating patient state at the time of an ECG, e.g., if a patient currently exhibits a left ventricular ejection fraction (LVEF) below 40%. *Patient forecasting* involves risk prediction over a future horizon, e.g., will a patient develop LVEF below 40% within one year of ECG acquisition.

In Section 5, we evaluate current ECG representations on exemplar tasks from the framework defined above. We find that performance can vary widely across task types. Our proposed taxonomy is a starting point rather than an exhaustive catalog of ECG applications. Future benchmarking should not be limited to (or necessarily include) them. We believe the community should come to a consensus on a set of clinically grounded tasks that belong in a standard ECG representation learning benchmark.

## 4. Toward Evaluation Best Practices

An evaluation protocol should *reliably stratify* representation quality, so that method rankings are robust to reasonable resampling and reporting choices. Unfortunately, much of the literature presents results in a way that obscures whether one representation is meaningfully better than another.

First, the field relies on macro-AUROC (Wu & Zhou, 2017) as its primary metric. In our survey, 89.29% of papers reported it as their headline metric. While convenient for evaluating multi-label data, macro-AUROC masks performance on the individual clinical tasks that clinicians care about. Furthermore, macro-averaging weights all labels equally. This implicitly grants rare, high-variance endpoints the same influence as common and more clinically salient ones, amplifying noise in reported rankings. However, 35.71% of papers did not report per-task performance.

Second, ECG benchmarks often include rare diagnoses that yield highly imbalanced labels with few positive examples. For example, under the standard PTB-XL protocol, 13 labels have fewer than 10 examples in the test set (Wagner et al., 2020; Strodthoff et al., 2021). In this case, per-label performance metrics have high sampling variability and can shift meaningfully because of small perturbations during resampling. Macro-AUROC inherits this variability, and can amplify it by giving equal weight to all labels.

Under extreme class imbalance, AUROC can remain deceptively high, even when a model yields low precision at clinically relevant operating points. When negative cases vastly outnumber positive cases, the false positive rate can remain small even when the absolute number of false positives is large. In this setting, AUPRC is an appropriate companion metric because it summarizes the tradeoff between precision and recall, thereby capturing whether the model can recover positive cases without producing an excessive number of false positives (Davis & Goadrich, 2006; Saito & Rehmsmeier, 2015).

AUROC and AUPRC are useful because they are threshold-independent summaries of predictive performance. How-

ever, they are not sufficient for many clinical use cases. In practice, relevant metrics depend on the intended application and operating point. For example, screening, confirmatory diagnosis, and patient forecasting typically prioritize different tradeoffs between false positives and false negatives. When a deployment setting is specified, evaluation should therefore also include appropriate threshold-dependent metrics, such as sensitivity, specificity, positive predictive value, and negative predictive value.

Lastly, uncertainty is rarely quantified in the field; 42.86% of papers did not report confidence intervals for their main results. As a result, many apparent gaps between methods are indistinguishable from sampling noise. Precise statistical testing is needed for claims of improvement over prior methods. The appropriate test depends on the experimental setting and methods.

We advocate for a standardized and statistically rigorous reporting protocol. The following practices are broadly applicable to multi-label benchmarks:

1. Treat macro-averaged metrics as a coarse summary of performance; report per-task AUROC and AUPRC.

2. Report bootstrapped confidence intervals, not only point estimates.

3. Use paired comparisons for claims of improvement over prior methods, for example, with paired bootstrap confidence intervals or statistical tests.

4. Exclude labels with insufficient number of test-set examples from quantitative evaluation.

ECG datasets have dozens of labels, so exhaustive reporting can be impractical in the main text of a work. However, these data should be available in the supplementary material. We recommend emphasizing the most clinically salient endpoints, for example, diagnosis of low ejection fraction. These core labels should be agreed upon by the community to enable meaningful comparison and mitigate cherry-picking results.

The importance of these recommendations is made transparent in Section 5 where we demonstrate that following them changes what one might conclude about the relative performance of current methods.

## 5. Empirical Study

### 5.1. Pre-training Configuration

**Models.** We evaluate downstream performance using representations from five pre-trained ECG encoders: CLOCS (Kiyasseh et al., 2021), KED (Tian et al., 2024), HeartLang (Jin et al., 2025), MERL (Liu et al., 2024b), and D-BETA

(Hung et al., 2024). As a baseline, we also evaluate embeddings from a randomly initialized 1D ResNet-18 encoder, a common architecture in ECG modeling (He et al., 2016; Ribeiro et al., 2020). To assess the sensitivity of this baseline to architectural and preprocessing choices, we evaluate a grid of randomly initialized encoders in Section 5.4.4.

**Pre-training Dataset.** We use the MIMIC-IV ECG database (Gow et al., 2023), which contains 800,035 10-second 12-lead ECGs collected from 161,352 patients. Each ECG is paired with a text diagnosis report.

**Implementation.** We use the publicly available MIMIC-IV checkpoints for KED, HeartLang, MERL and D-BETA. For CLOCS, we retrain on MIMIC-IV following the authors' training procedures so all models use the same pre-training corpus; this isolates differences in model design and objective rather than pre-training data. Full pre-training details are provided in Appendix C.2. All experiments are conducted on one NVIDIA Tesla V100-SXM2-32GB GPU.

### 5.2. Downstream Tasks

We evaluate all ECG encoders with linear probing on six downstream settings. Full dataset details, including label definitions and prevalence, are provided in Appendix B.

All ECGs are standardized to 10-second 12-lead segments in millivolts sampled at 500 Hz. We remove recordings with `NaN` or `Inf` samples. We use dataset-level train/val/test splits (70/10/20), except PTB-XL and EchoNext, which use standard splits, and the patient forecasting task, which follows a 75/10/15 split (Poterucha et al., 2025; Strodthoff et al., 2021; Bergamaschi et al., 2025).

**PTB-XL.** PTB-XL consists of 21,837 12-lead 10-second ECGs from 18,885 patients (Wagner et al., 2022). It is split into four multi-label classification tasks that assess arrhythmia and waveform abnormalities, with varying numbers of binary targets: SUPER (5 labels), SUB (23 labels), FORM (19 labels), and RHYTHM (12 labels). Each task has a different number of samples, as detailed in Appendix B.1.

**CPSC2018.** This dataset includes 6,877 12-lead ECGs, with arrhythmia and waveform morphology annotations (Liu et al., 2018). Recording duration varies between 5 and 72 seconds. We exclude recordings shorter than 10 seconds, and for longer recordings, clip them to 10 seconds. Appendix B.2 contains more details.

**CSN.** The Chapman-Shaoxing-Ningbo (CSN) database includes 45,152 10-second 12-lead ECGs from 10,646 patients (Zheng et al., 2022). ECGs are annotated with arrhythmia and waveform morphology labels; see Appendix B.3.

**EchoNext.** For binary classification of structural heart disease from the ECG, we use EchoNext, a dataset of 100,000 10-second 12-lead ECGs (Poterucha et al., 2025). Each ECG

*Table 1.* On the standard arrhythmia/waveform benchmarks, KED and D-BETA appear to dominate according to macro-AUROC. However, once uncertainty is quantified, neither ECG pre-training method consistently prevails. A randomly initialized encoder is often competitive, often beating the ECG-specific pre-training methods CLOCS and HeartLang. Entries report macro-AUROC with 95% confidence intervals in the subscript under linear probing at varying levels of training data. Green indicates no significant difference from the top method.

| Dataset | | Random | CLOCS | KED | HeartLang | MERL | D-BETA |
|---|---|---|---|---|---|---|---|
| PTB-XL | 1% | 0.770 $_{0.759–0.780}$ | 0.752 $_{0.740–0.764}$ | 0.861 $_{0.852–0.869}$ | 0.645 $_{0.632–0.658}$ | 0.823 $_{0.813–0.832}$ | 0.867 $_{0.858–0.875}$ |
| SUPER | 10% | 0.832 $_{0.822–0.842}$ | 0.807 $_{0.796–0.817}$ | 0.894 $_{0.886–0.901}$ | 0.790 $_{0.780–0.800}$ | 0.884 $_{0.876–0.892}$ | 0.887 $_{0.879–0.895}$ |
| | 100% | 0.861 $_{0.851–0.870}$ | 0.821 $_{0.810–0.830}$ | 0.909 $_{0.902–0.915}$ | 0.824 $_{0.815–0.834}$ | 0.901 $_{0.893–0.908}$ | 0.893 $_{0.885–0.901}$ |
| PTB-XL | 1% | 0.689 $_{0.673–0.704}$ | 0.657 $_{0.626–0.688}$ | 0.783 $_{0.770–0.795}$ | 0.574 $_{0.548–0.599}$ | 0.757 $_{0.740–0.772}$ | 0.791 $_{0.781–0.802}$ |
| SUB | 10% | 0.763 $_{0.739–0.786}$ | 0.771 $_{0.752–0.790}$ | 0.878 $_{0.857–0.898}$ | 0.708 $_{0.688–0.729}$ | 0.857 $_{0.837–0.875}$ | 0.862 $_{0.834–0.888}$ |
| | 100% | 0.834 $_{0.808–0.859}$ | 0.803 $_{0.784–0.819}$ | 0.920 $_{0.909–0.929}$ | 0.836 $_{0.819–0.853}$ | 0.905 $_{0.890–0.918}$ | 0.899 $_{0.882–0.914}$ |
| PTB-XL | 1% | 0.542 $_{0.528–0.557}$ | 0.562 $_{0.546–0.577}$ | 0.656 $_{0.643–0.670}$ | 0.526 $_{0.510–0.542}$ | 0.620 $_{0.607–0.635}$ | 0.657 $_{0.643–0.670}$ |
| FORM | 10% | 0.684 $_{0.658–0.709}$ | 0.655 $_{0.631–0.679}$ | 0.719 $_{0.688–0.753}$ | 0.570 $_{0.547–0.594}$ | 0.735 $_{0.713–0.757}$ | 0.763 $_{0.738–0.787}$ |
| | 100% | 0.756 $_{0.734–0.780}$ | 0.715 $_{0.687–0.741}$ | 0.851 $_{0.827–0.875}$ | 0.707 $_{0.679–0.736}$ | 0.853 $_{0.839–0.866}$ | 0.845 $_{0.821–0.868}$ |
| PTB-XL | 1% | 0.499 $_{0.474–0.526}$ | 0.708 $_{0.689–0.728}$ | 0.810 $_{0.789–0.831}$ | 0.569 $_{0.505–0.630}$ | 0.746 $_{0.708–0.778}$ | 0.834 $_{0.812–0.854}$ |
| RHYTHM | 10% | 0.776 $_{0.749–0.804}$ | 0.802 $_{0.777–0.827}$ | 0.940 $_{0.926–0.952}$ | 0.731 $_{0.679–0.785}$ | 0.878 $_{0.847–0.905}$ | 0.956 $_{0.941–0.968}$ |
| | 100% | 0.787 $_{0.737–0.833}$ | 0.810 $_{0.762–0.854}$ | 0.959 $_{0.948–0.969}$ | 0.854 $_{0.827–0.879}$ | 0.903 $_{0.870–0.933}$ | 0.968 $_{0.956–0.978}$ |
| CPSC2018 | 1% | 0.630 $_{0.615–0.645}$ | 0.696 $_{0.677–0.715}$ | 0.855 $_{0.843–0.867}$ | 0.615 $_{0.599–0.632}$ | 0.835 $_{0.822–0.849}$ | 0.926 $_{0.916–0.937}$ |
| | 10% | 0.779 $_{0.763–0.794}$ | 0.769 $_{0.753–0.784}$ | 0.923 $_{0.914–0.932}$ | 0.719 $_{0.702–0.736}$ | 0.887 $_{0.873–0.900}$ | 0.942 $_{0.933–0.951}$ |
| | 100% | 0.850 $_{0.834–0.864}$ | 0.802 $_{0.787–0.816}$ | 0.952 $_{0.945–0.958}$ | 0.849 $_{0.836–0.862}$ | 0.927 $_{0.916–0.937}$ | 0.958 $_{0.951–0.966}$ |
| CSN | 1% | 0.603 $_{0.597–0.609}$ | 0.620 $_{0.614–0.626}$ | 0.695 $_{0.689–0.700}$ | 0.553 $_{0.548–0.560}$ | 0.663 $_{0.658–0.668}$ | 0.726 $_{0.722–0.730}$ |
| | 10% | 0.710 $_{0.695–0.724}$ | 0.734 $_{0.716–0.754}$ | 0.832 $_{0.819–0.845}$ | 0.686 $_{0.672–0.700}$ | 0.792 $_{0.772–0.811}$ | 0.860 $_{0.850–0.868}$ |
| | 100% | 0.763 $_{0.741–0.785}$ | 0.809 $_{0.792–0.826}$ | 0.910 $_{0.900–0.919}$ | 0.791 $_{0.776–0.806}$ | 0.862 $_{0.853–0.872}$ | 0.947 $_{0.941–0.952}$ |
| ECHONEXT | 1% | 0.694 $_{0.678–0.707}$ | 0.621 $_{0.607–0.636}$ | 0.687 $_{0.671–0.704}$ | 0.637 $_{0.622–0.652}$ | 0.691 $_{0.677–0.706}$ | 0.692 $_{0.678–0.708}$ |
| | 10% | 0.751 $_{0.737–0.764}$ | 0.702 $_{0.690–0.713}$ | 0.730 $_{0.714–0.745}$ | 0.704 $_{0.691–0.717}$ | 0.768 $_{0.753–0.782}$ | 0.754 $_{0.742–0.766}$ |
| | 100% | 0.773 $_{0.761–0.784}$ | 0.714 $_{0.702–0.725}$ | 0.766 $_{0.752–0.779}$ | 0.736 $_{0.723–0.750}$ | 0.791 $_{0.780–0.801}$ | 0.774 $_{0.763–0.784}$ |

was paired with a contemporaneous echocardiogram, from which structural heart disease labels were derived. Details are in Appendix B.4.

**Hemodynamic Inference.** We use a private dataset of 9,226 10-second 12-lead ECGs from 5,072 patients at Massachusetts General Hospital (MGH) (Schlesinger et al., 2022). We consider two diagnosis tasks: contemporaneous mean pulmonary capillary wedge pressure (mPCWP) and mean pulmonary artery pressure (mPA), each measured by ground-truth right heart catheterization. Cohort construction and labeling are described in Appendix B.5.

**Patient Forecasting.** We consider the binary prediction task of whether a patient will experience heart failure within one year of an ECG (1YR-HF). We define heart failure as echocardiographic left ventricular ejection fraction below 40%. We use the private dataset of (Bergamaschi et al., 2025), which includes 913,420 10-second 12-lead ECGs from 82,244 patients at MGH. See Section B.6.

### 5.3. Evaluation Protocol

We evaluate the downstream performance of each representation using linear probing. For each individual task, we freeze the ECG encoder and train a single $\ell_2$-regularized logistic regression model on top of the embeddings using the training set. We run a hyperparameter sweep, detailed in Appendix C.1, pick the best probe for each task based on the validation set, then evaluate that probe on the test set.

We report the AUROC and AUPRC for each individual task. We additionally aggregate over tasks to report the macro-AUROC for each dataset. To quantify uncertainty, for all experiments, we conduct a paired bootstrap with 1,000 resamples with replacement on the test set and report 95% confidence intervals.

In many of the following tables, we highlight methods whose performance is not significantly different from that of the top-performing method. To make this determination, for

*Table 2.* Task-level results reveal heterogeneous behavior. Shown are the five highest-prevalence PTB-XL SUB labels plus CLBBB. Method rankings vary by endpoint. AUPRC can help capture differences masked by AUROC, as in the case of CLBBB. Each cell reports AUROC (top) and AUPRC (bottom) with 95% confidence intervals; methods not statistically different from the best are highlighted in green. Results on the remaining tasks are in Table 20.

| Method | NORM | IMI | AMI | STTC | LVH | CLBBB |
|---|---|---|---|---|---|---|
| Random | 0.891 0.877–0.903 / 0.839 0.813–0.863 | 0.864 0.840–0.884 / 0.556 0.506–0.609 | 0.894 0.874–0.915 / 0.646 0.593–0.700 | 0.827 0.801–0.851 / 0.319 0.275–0.368 | 0.917 0.896–0.935 / 0.648 0.593–0.704 | 0.973 0.934–0.999 / 0.880 0.784–0.955 |
| CLOCS | 0.859 0.843–0.872 / 0.788 0.763–0.812 | 0.674 0.644–0.702 / 0.299 0.262–0.341 | 0.857 0.832–0.883 / 0.590 0.537–0.646 | 0.823 0.796–0.849 / 0.319 0.279–0.366 | 0.900 0.875–0.921 / 0.593 0.528–0.650 | 0.984 0.974–0.993 / 0.676 0.551–0.794 |
| KED | 0.932 0.921–0.942 / 0.900 0.883–0.916 | 0.881 0.862–0.899 / 0.597 0.549–0.647 | 0.951 0.939–0.962 / 0.808 0.770–0.845 | 0.893 0.873–0.911 / 0.507 0.444–0.570 | 0.949 0.935–0.961 / 0.745 0.695–0.794 | 0.998 0.997–1.000 / 0.949 0.906–0.983 |
| HeartLang | 0.875 0.860–0.890 / 0.822 0.798–0.846 | 0.750 0.721–0.776 / 0.353 0.310–0.399 | 0.876 0.857–0.895 / 0.548 0.496–0.601 | 0.799 0.772–0.825 / 0.295 0.250–0.346 | 0.823 0.794–0.853 / 0.405 0.343–0.472 | 0.993 0.984–0.998 / 0.867 0.767–0.944 |
| MERL | 0.926 0.915–0.937 / 0.885 0.862–0.904 | 0.843 0.819–0.864 / 0.538 0.488–0.589 | 0.959 0.949–0.969 / 0.841 0.809–0.873 | 0.885 0.863–0.905 / 0.480 0.421–0.539 | 0.929 0.912–0.945 / 0.680 0.623–0.734 | 0.999 0.997–1.000 / 0.947 0.889–0.988 |
| D-BETA | 0.929 0.919–0.939 / 0.894 0.875–0.912 | 0.897 0.880–0.914 / 0.691 0.649–0.734 | 0.956 0.943–0.967 / 0.834 0.793–0.869 | 0.862 0.837–0.887 / 0.429 0.370–0.490 | 0.862 0.837–0.885 / 0.489 0.426–0.552 | 0.999 0.997–1.000 / 0.970 0.937–0.995 |

*Table 3.* Tasks with very few positives can yield high-variance estimates that distort benchmark summaries. Confidence intervals are wide for two of the three tasks with lowest prevalence in PTB-XL FORM. Each cell reports AUROC (top) and AUPRC (bottom) with 95% confidence intervals.

| Method | PRC(S) | STE_ | TAB_ |
|---|---|---|---|
| Random | 0.92 0.91–0.94 / 0.02 0.01–0.02 | 0.64 0.41–0.78 / 0.01 0.01–0.01 | 0.56 0.39–0.84 / 0.01 0.00–0.02 |
| CLOCS | 0.85 0.83–0.87 / 0.01 0.01–0.01 | 0.56 0.10–0.91 / 0.01 0.00–0.04 | 0.70 0.59–0.80 / 0.01 0.01–0.02 |
| KED | 0.97 0.96–0.98 / 0.03 0.03–0.05 | 0.59 0.30–0.94 / 0.01 0.00–0.05 | 0.74 0.52–0.98 / 0.03 0.01–0.13 |
| HeartLang | 0.87 0.85–0.89 / 0.01 0.01–0.01 | 0.61 0.20–0.93 / 0.01 0.00–0.05 | 0.27 0.14–0.50 / 0.00 0.00–0.01 |
| MERL | 0.76 0.73–0.79 / 0.01 0.00–0.01 | 0.92 0.84–0.99 / 0.11 0.02–0.43 | 0.87 0.80–0.97 / 0.04 0.01–0.12 |
| D-BETA | 0.97 0.96–0.98 / 0.04 0.03–0.06 | 0.71 0.51–0.95 / 0.02 0.01–0.06 | 0.66 0.23–0.95 / 0.02 0.00–0.07 |

each task, we first identify the best-performing method according to each metric. We then compare each other method against it using a two-sided paired permutation test with 1,000 permutation replicates. A method is considered not significantly different from the top-performing method if its Bonferroni-corrected p-value exceeds 0.05 on any metric. We note that this choice of test is not universally optimal, and the most appropriate statistical procedure depends on the experimental setting and the methods under comparison.

To assess performance in a limited-label regime, on PTB-XL, CPSC2018, CSN, and EchoNext, we repeat this probing procedure using 1%, 10%, and 100% of the available labeled training data.

## 5.4. Experimental Results

### 5.4.1. EVALUATION ON PTB-XL, CPSC2018, CSN

**Macro Performance.** Table 1 shows that conclusions drawn from macro-AUROC can change substantially once a random encoder baseline and uncertainty are included. KED and D-BETA often have the highest reported macro-AUROC, but the apparent winner changes once uncertainty is considered. For example, KED and D-BETA are often statistically indistinguishable, and MERL matches the top-performing method on PTB-XL FORM at 10% and 100% data and on PTB-XL SUB at 100% data. The randomly initialized encoder is competitive, frequently exceeding CLOCS and HeartLang, including on PTB-XL SUPER at all training fractions and on CPSC2018 at 10% and 100% of the data. We revisit the robustness of this observation in Section 5.4.4. Overall, many differences between methods fall within statistical noise, indicating that rankings based solely on macro-AUROC estimates can be unreliable.

**Task-level Performance.** We next examine task-level performance within each dataset. Full results are available in Appendices D.1, E, and F. We report results for the five labels with highest prevalence from PTB-XL SUB in Table 2 and include complete left bundle branch block (CLBBB) as an illustrative example. The randomly initialized encoder is consistently competitive with CLOCS and HeartLang, and is among the best-performing within sampling noise on CLBBB. KED, MERL, and D-BETA provide gains on some tasks (e.g., STTC), but their relative ranking depends on the endpoint. Finally, CLBBB illustrates why AUROC alone can be misleading: AUROC is near-saturated for all methods, whereas AUPRC reveals better separation, with D-BETA clearly outperforming CLOCS, HeartLang, and the random baseline, while MERL and KED are intermediate.

*Table 4.* Removing low-support labels can materially change macro-AUROC and alter which method appears most performant. ORIG uses the standard PTB-XL FORM test set; CLEAN excludes labels with fewer than 10 positive test examples. Values are macro-AUROC with 95% confidence intervals.

| Method | ORIG | CLEAN | Δ |
|---|---|---|---|
| RANDOM | 0.756 0.734–0.780 | 0.754 0.733–0.774 | -0.002 |
| CLOCS | 0.715 0.687–0.741 | 0.710 0.690–0.731 | -0.005 |
| KED | 0.851 0.827–0.875 | 0.862 0.846–0.876 | +0.011 |
| HEARTLANG | 0.707 0.679–0.736 | 0.723 0.699–0.745 | +0.016 |
| MERL | 0.853 0.839–0.866 | 0.847 0.833–0.860 | -0.005 |
| D-BETA | 0.845 0.821–0.868 | 0.855 0.841–0.868 | +0.009 |

*Table 5.* On structural disease endpoints, CLOCS and HeartLang underperform while the other methods cluster closely. Each cell reports AUROC/AUPRC with 95% confidence intervals; methods not significantly different from the top-performing method are highlighted in green.

| Method | SHD | LVEF $\leq 45$ | TR |
|---|---|---|---|
| Random | 0.81 0.79–0.82 / 0.77 0.76–0.79 | 0.86 0.84–0.87 / 0.62 0.59–0.65 | 0.78 0.76–0.81 / 0.23 0.20–0.27 |
| CLOCS | 0.72 0.71–0.74 / 0.69 0.67–0.70 | 0.78 0.76–0.79 / 0.48 0.45–0.51 | 0.71 0.69–0.74 / 0.14 0.12–0.17 |
| KED | 0.80 0.79–0.81 / 0.76 0.75–0.78 | 0.86 0.85–0.87 / 0.62 0.59–0.65 | 0.78 0.75–0.80 / 0.23 0.19–0.26 |
| HeartLang | 0.77 0.76–0.79 / 0.72 0.70–0.74 | 0.83 0.81–0.84 / 0.55 0.52–0.58 | 0.75 0.73–0.78 / 0.18 0.16–0.21 |
| MERL | 0.81 0.80–0.82 / 0.78 0.76–0.80 | 0.88 0.87–0.89 / 0.67 0.64–0.70 | 0.82 0.79–0.84 / 0.27 0.23–0.30 |
| D-BETA | 0.80 0.79–0.81 / 0.76 0.75–0.78 | 0.86 0.85–0.87 / 0.61 0.58–0.64 | 0.79 0.77–0.82 / 0.24 0.20–0.27 |

### 5.4.2. SENSITIVITY OF MACRO-AVERAGED METRICS TO LABELS WITH FEW EXAMPLES

In Section 4, we claim that the current practice of retaining labels with low test-support leads to noisy performance metrics. To illustrate this point, Table 3 highlights the AUROC for the three tasks with the lowest prevalence in PTB-XL FORM. Each task has small test-support (PRC(S): 1 positive, STE_: 3 positives, TAB_: 3 positives), which can produce very wide confidence intervals and highly variable AUROC estimates across resamples. This variance has a material effect on the resulting macro-AUROC for each model. To demonstrate this, when we remove the PTB-XL FORM labels with fewer than 10 positive test examples (4 labels in total), the resulting macro-AUROC can shift non-trivially. This is shown in Table 4, where there is a change in the apparent ordering of the strongest methods: MERL is highest under the original label set, whereas KED is highest after low-support labels are removed. Parallel results for PTB-XL SUB and RHYTHM are provided in Appendix D.2.

*Table 6.* The competitiveness of the random baseline in this paper is not an artifact of a single configuration. Values are macro-AUROC with standard deviation across grid of randomly initialized encoders. ALL summarizes all 36 random encoder configurations; RESNET and VIT summarize each backbone subset.

| Dataset | ALL | RESNET | VIT |
|---|---|---|---|
| PTB-XL SUPER | 0.82 ± 0.07 | 0.87 ± 0.01 | 0.76 ± 0.07 |
| PTB-XL SUB | 0.79 ± 0.09 | 0.86 ± 0.02 | 0.71 ± 0.08 |
| PTB-XL FORM | 0.70 ± 0.07 | 0.76 ± 0.01 | 0.64 ± 0.05 |
| PTB-XL RHYTHM | 0.75 ± 0.09 | 0.81 ± 0.03 | 0.68 ± 0.07 |
| CPSC2018 | 0.79 ± 0.09 | 0.86 ± 0.01 | 0.73 ± 0.08 |
| CSN | 0.76 ± 0.09 | 0.82 ± 0.03 | 0.69 ± 0.07 |
| ECHONEXT | 0.75 ± 0.03 | 0.77 ± 0.01 | 0.73 ± 0.03 |

### 5.4.3. EVALUATION ON ECHONEXT

We consider EchoNext to illustrate how conclusions about method quality differ on structural disease endpoints. CLOCS and HeartLang underperform when the linear probe is trained with 1%, 10%, or 100% of the available training data (Table 1). At 1% of the data, the random baseline has the highest macro-AUROC, but falls within sampling noise of KED, MERL, and D-BETA. At 100% data, MERL achieves the strongest macro-AUROC, with the random baseline close behind. We further report performance on three clinically important structural endpoints (Table 5); the same pattern holds at the task level, with CLOCS and HeartLang consistently worse and the other methods tightly clustered. Additional results are available in Appendix G.

### 5.4.4. ROBUSTNESS OF RANDOM ENCODER BASELINE

To characterize the robustness of the randomly initialized baseline, we reran our evaluation using 36 additional random encoders. Each encoder used a different combination of ECG sampling rate (100, 250, or 500 Hz), filtering method (none or order-5 Butterworth bandpass of 0.5 to 40 Hz), normalization method (none, dataset-level z-score, or per-sample z-score), and backbone (ResNet18 or ViT-Medium (Dosovitskiy et al., 2021)). Our preprocessing steps were based on common configurations in the ECG modeling literature. For each encoder, we applied the same linear probing procedure described in Section 5.3.

Table 6 shows the mean macro-AUROC across this grid at 100% label availability on each public dataset. We additionally report the performance within the ResNet and ViT subsets. Standard deviation is reported across encoders. Unsurprisingly, the backbone affects performance, as ResNet18 outperformed ViT-Medium. However, while the performance of the random baseline is setup dependent, its competitiveness is not an artifact of the original configuration, whose results are in Table 1. Across a broad grid of reasonable choices it remains a non-trivial baseline, and is often competitive with at least some of the pre-trained methods.

*Table 7.* Performance on mPCWP, mPA, and 1YR-HF (AU-ROC/AUPRC; 95% CIs). Most methods are statistically similar on the hemodynamic tasks; MERL and KED prevail on 1YR-HF.

| Method | mPCWP | mPA | 1YR-HF |
|---|---|---|---|
| Random | 0.70 $_{0.67-0.73}$ / 0.71 $_{0.68-0.75}$ | 0.72 $_{0.68-0.76}$ / 0.86 $_{0.83-0.88}$ | 0.78 $_{0.78-0.79}$ / 0.58 $_{0.58-0.59}$ |
| CLOCS | 0.68 $_{0.64-0.71}$ / 0.71 $_{0.68-0.74}$ | 0.66 $_{0.63-0.70}$ / 0.84 $_{0.81-0.86}$ | 0.75 $_{0.74-0.75}$ / 0.53 $_{0.53-0.54}$ |
| KED | 0.72 $_{0.69-0.75}$ / 0.75 $_{0.71-0.78}$ | 0.76 $_{0.73-0.79}$ / 0.89 $_{0.87-0.91}$ | 0.83 $_{0.82-0.83}$ / 0.66 $_{0.65-0.66}$ |
| HeartLang | 0.68 $_{0.65-0.71}$ / 0.70 $_{0.66-0.73}$ | 0.71 $_{0.67-0.74}$ / 0.86 $_{0.84-0.88}$ | 0.79 $_{0.78-0.79}$ / 0.58 $_{0.58-0.59}$ |
| MERL | 0.74 $_{0.71-0.77}$ / 0.76 $_{0.73-0.79}$ | 0.76 $_{0.73-0.80}$ / 0.88 $_{0.86-0.90}$ | 0.83 $_{0.83-0.83}$ / 0.66 $_{0.66-0.67}$ |
| D-BETA | 0.71 $_{0.68-0.74}$ / 0.73 $_{0.70-0.77}$ | 0.74 $_{0.71-0.78}$ / 0.87 $_{0.85-0.90}$ | 0.82 $_{0.82-0.82}$ / 0.64 $_{0.63-0.64}$ |

### 5.4.5. HEMODYNAMIC INFERENCE

Table 7 shows results for the two hemodynamics tasks. MERL achieves the highest AUROC on mPCWP, while MERL and KED tie for best on mPA. Performance differences across methods appear to be smaller than on the standard public benchmarks. All except for HeartLang are within statistical noise of MERL on mPCWP; D-BETA is within statistical noise of MERL and KED on mPA, with the randomly initialized encoder close behind.

### 5.4.6. PATIENT FORECASTING

On 1-year heart failure forecasting (Table 7), MERL and KED are best with D-BETA following closely, all outperforming Random, CLOCS, and HeartLang on AUROC and AUPRC. Random also exceeds CLOCS ($\Delta$AUROC = 0.03; $\Delta$AUPRC = 0.05), reinforcing that a randomly initialized baseline is non-trivial even for patient-level forecasting.

## 6. Alternative Views

In Section 3 we argue that current downstream evaluation is narrow and should expand to include other clinical targets, such as structural disease, hemodynamic state, and forecasting tasks. A natural concern is that some of these targets cannot admit near-perfect performance due to aleatoric uncertainty, rendering them ill-suited for benchmarking (Ghassemi et al., 2020; Kohane et al., 2021; Pillai et al., 2024; Yuan et al., 2021). Even so, many influential ML benchmarks have also remained far from saturation, often due to noise and ambiguity, yet have driven progress by rewarding better representations and modeling choices (Northcutt et al., 2021; Vendrow et al., 2025). When an endpoint cannot be perfectly inferred from an ECG, the signal that *is* present

can be clinically meaningful and transferable toward other objectives. Hence, it is important that ECG representations are optimized to capture this information.

We also argue that ECG representations should be evaluated on a broad range of clinical tasks, ideally with standardized and transparent benchmarks. However, many valuable ECG endpoints exist in health systems with non-trivial barriers to public release. Some might therefore object that such endpoints should be excluded from benchmarking because private evaluations are less reproducible. We disagree. There is broad precedent in the ML community for benchmarking on hidden test sets, where the data are not released (e.g., Geiger et al., 2012; Wang et al., 2018; Perez Alday et al., 2022). Modern infrastructure enables containerized evaluation where benchmark hosts can run inference on behalf of a participant (Pavao et al., 2023). Private-endpoint benchmarks are especially appropriate for ECG representation learning, where downstream evaluation often tests transferability to tasks unseen during training. Hosts should publish detailed documentation including cohort and label construction, and provide a transparent auditing pathway when possible. Furthermore, expanding public resources for such tasks should be a priority.

## 7. Discussion

This paper analyzed evaluation of 12-lead ECG representations. We proposed a taxonomy of clinically grounded tasks, then outlined best practices that the field often fails to follow. Our experiments show that ranking current methods is difficult in practice, and that randomly initialized encoders are surprisingly strong baselines. When random encoders are competitive with pre-trained encoders, gains from pre-training should be interpreted with care, especially if they are small in absolute terms. Future work should explore why the random encoder is so performant for many ECG tasks. We hypothesize that the clinically relevant signal is so apparent that random convolutions do not distort it.

One limitation of our study is that we focused only on 12-lead ECG representation learning. We expect that many of our general conclusions hold for pre-trained encoders of other bio-signals (e.g., 1-lead ECG, PPG, and EEG), task-specific ECG models, and multi-modal representations that include ECG as a component (e.g., Radhakrishnan et al., 2023; Thapa et al., 2024); evaluating such models should be addressed in future work.

The field of ECG analysis would benefit from an extensible open-source framework with standardized evaluation code. The community should agree on a core set of clinically important tasks that will drive the future of ECG pre-training. Our hope is that by doing so, the field will converge on broadly useful and generalizable representations.

## Acknowledgements

We thank Tiffany Yau for guidance with the hemodynamic inference and patient forecasting tasks. We also thank Danielle Pace and Roey Ringel for helpful discussions and feedback. Zachary Berger is supported by the Department of Defense NDSEG Fellowship. This work was also supported by Quanta Computer Inc.

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

# A. Selection of Model Survey Set

We construct a *model survey set* of papers that propose a representation learning method for 12-lead ECGs. In the literature, these are commonly referred to as pre-trained ECG encoders or ECG foundation models.

We consider papers published between January 1, 2019 and December 31, 2025. We begin our survey in 2019 since that date coincides with the invention of contemporary self-supervised and large-scale pretraining methods applied to ECGs, e.g. SimCLR and BYOL (Chen et al., 2020; Grill et al., 2020).

To form the survey set, we first identified methods published in top ML and ML-health venues: ICML, ICLR, NeurIPS, AAAI, CHIL, and ML4H. For each venue, we required that one keyword from each of the following sets appeared in the paper title or abstract:

**Cardiac Keyword.** ECG, EKG, electrocardiogram, electrocardiography, cardiac, 12-lead, multi-lead, multilead.

**Representation Learning Keyword.** SSL, self-supervised, contrastive, masked, reconstruction, reconstructive, foundation model, pretrain, pre-train, pre-training, pretraining, multimodal, multi-modal, representation.

If a paper plausibly proposed a transferable 12-lead ECG representation based on its title or abstract, we screened the full text. A paper was included in the survey set if it met the following inclusion criteria:

1. 12-lead ECG is one of the primary modalities considered.

2. The paper proposes a method designed to produce a reusable representation.

3. The paper reports downstream evaluation on at least one 12-lead ECG task using linear probing or fine-tuning.

Because a substantial amount of ECG representation learning work appears outside this ecosystem, we then added any method cited by the initially identified papers in their introduction, related works or as a baseline, provided they fit our inclusion criteria. While this procedure may have missed some relevant papers, e.g., some included in (Han et al., 2025), it has the desirable property of restricting our discussion to work that is directly pertinent to the ML-methods community.

The initial search from the ML venues returned 56 papers, with 11 papers left after screening. There were then an additional 17 cited papers that were included. The resulting survey set contains 28 papers, listed in Table 8. We use this set when making claims about common benchmarking and reporting practices in 12-lead ECG representation learning.

For each method in the survey set, we extracted the following data, which is summarized for each method in Table 9:

- Publication venue.

- Year of publication.

- Pre-training dataset used to develop the representation.

- Downstream datasets the representation is evaluated on.

- Which performance metrics were reported.

- If task-level metrics were reported, or only macro-averaged metrics.

- Whether uncertainty was quantified in the paper.

- If the method compares to a random encoder as a baseline.

*Table 8.* Selected survey set of 12-lead ECG representation learning methods. For each method we indicate the year and venue of publication, which datasets were used for pre-training, and which datasets were used for downstream evaluation. PhysioNet 2020 includes PTB-XL, CPSC2018, INCART, and G12EC. PhysioNet 2021 includes PTB-XL, CPSC2018, INCART, and G12EC, CSN, and UMich.

| Method | Year | Venue | Pretraining Dataset | Evaluation Dataset |
|---|---|---|---|---|
| CLOCS (Kiyasseh et al., 2021) | 2021 | ICML | PhysioNet 2020, Chapman | PhysioNet 2020, Chapman, Cardiology, PhysioNet 2017 |
| 3KG (Gopal et al., 2021) | 2021 | ML4H | PhysioNet 2020 | PhysioNet 2020 |
| ISL (Lan et al., 2022) | 2022 | AAAI | PTB-XL, Chapman, CPSC2018 | PTB-XL, Chapman, CPSC2018 |
| (Oh et al., 2022) | 2022 | CHIL | PTB-XL, CPSC2018, G12EC, CSN | PTB-XL, CPSC2018, G12EC |
| CRT (Zhang et al., 2024) | 2022 | TNNLS | PTB-XL, HAR, Sleep-EDF | PTB-XL, HAR, Sleep-EDF |
| CPC (Mehari & Strodthoff, 2022) | 2022 | CIBM | PhysioNet 2020, Chapman, Ribeiro | PTB-XL |
| PCLR (Diamant et al., 2022) | 2022 | PLOS-CB | Private Dataset | Private Dataset |
| CT-HB (Wei et al., 2022) | 2022 | ICASSP | MIT-BIH, Chapman | MIT-BIH, Chapman, Private Dataset |
| BIOT (Yang et al., 2023) | 2023 | NeurIPS | SHHS, PREST, PhysioNet 2020 | PTB-XL, CHB-MIT, IIIC Seizure, TUAB, TUEV, HAR |
| ASTCL (Wang et al., 2024) | 2023 | TNNLS | PTB-XL, Chapman, CODE, CPSC2018, CMI | PTB-XL, Chapman, CODE, CPSC2018, CMI |
| sEHR-ECG (Lalam et al., 2023) | 2023 | TMLR | Private Dataset | PhysioNet 2020, Chapman, Private Dataset |
| (Lai et al., 2023) | 2023 | Nat. Comms. | Private Dataset | Private Dataset, CPSC2018 |
| METS (Li et al., 2023) | 2023 | MIDL | PTB-XL | PTB-XL, MIT-BIH |
| MaeFE (Zhang et al., 2023a) | 2023 | IEEETIM | CPSC2018, Ningbo | PTB-XL, CPSC2018 |
| sCL-ST (Le et al., 2023) | 2023 | IEEEJBHI | CPSC2018, INCART, G12EC, PTB | PTB-XL |
| T-S Reverse (Zhang et al., 2023b) | 2023 | BSPC | PhysioNet 2017 | PhysioNet 2017 |
| ST-MEM (Na et al., 2024) | 2024 | ICLR | CSN, CODE-15 | PTB-XL, CPSC2018, PhysioNet 2017 |
| ESI (Yu et al., 2024) | 2024 | TMLR | PTB-XL, MIMIC-IV-ECG, Chapman | PTB-XL, ICBEB |
| ETP (Liu et al., 2024a) | 2024 | ICASSP | PTB-XL | PTB-XL and CPSC2018 |
| KED (Tian et al., 2024) | 2024 | Cell-RM | MIMIC-IV-ECG | CPSC2018, Chapman, G12EC, PTB-XL, Private Dataset |
| MERL (Liu et al., 2024b) | 2024 | ICML | MIMIC-IV | PTB-XL, CPSC2018, CSN |
| D-BETA (Hung et al., 2024) | 2025 | ICML | MIMIC-IV | PhysioNet 2021, CODE-test |
| MELP (Wang et al., 2025) | 2025 | ICML | MIMIC-IV | PTB-XL, CPSC2018, CSN |
| H-Tuning (Zhou et al., 2025) | 2025 | ICML | CODE | PTB-XL, CSN, G12EC, Private Dataset |
| HeartLang (Jin et al., 2025) | 2025 | ICLR | MIMIC-IV-ECG | PTB-XL, CPSC2018, and Chapman |
| ECG-FM (McKeen et al., 2025) | 2025 | JAMIA Open | CPSC2018, PTB-XL, G12EC, CSN, MIMIC-IV | UHN-ECG, MIMIC-IV |
| ECGFounder (Li et al., 2025) | 2025 | NEJM-AI | Harvard-Emory ECG Database | PTB-XL, CODE-test, PhysioNet 2017, MIMIC-IV, Private Dataset |
| CREMA (Song et al., 2025) | 2025 | CIKM | MIMIC-IV, CODE-15, UKBB, SaMi-Trop, IKEM | PTB-XL |

*Table 9.* Evaluation and reporting choices in our surveyed set of 12-lead ECG representation learning papers. "Y" indicates the practice is clearly reported, and "N" otherwise. The column *per-label* indicates whether the given method reports its results per individual task, or aggregates them together. *UQ* indicates whether the paper uses uncertainty quantification when reporting their results. *Rand* indicates whether the paper compares to a random encoder as a baseline. *AUROC* denotes "area under the receiver operating curve", *AUPRC* denotes "area under the precision-recall curve", and *Acc.* denotes "accuracy".

| Method | Metrics Reported | Per-label? | UQ? | Rand? |
|---|---|---|---|---|
| CLOCS (Kiyasseh et al., 2021) | AUROC | N | Y | Y |
| 3KG (Gopal et al., 2021) | AUROC, $F_1$ | Y | Y | N |
| ISL (Lan et al., 2022) | AUROC | N | Y | Y |
| (Oh et al., 2022) | Acc. | N | Y | Y |
| CRT (Zhang et al., 2024) | AUROC, Acc., $F_1$ | Y | Y | N |
| CPC (Mehari & Strodthoff, 2022) | AUROC | Y | Y | N |
| PCLR (Diamant et al., 2022) | $F_1$, $R^2$ | Y | Y | N |
| CT-HB (Wei et al., 2022) | AUROC, Acc., MCC, Sensitivity, Specificity, PPV | Y | N | N |
| BIOT (Yang et al., 2023) | AUROC, Acc., AUPRC, $F_1$ | N | Y | N |
| ASTCL (Wang et al., 2024) | AUROC, $F_1$ | Y | Y | Y |
| sEHR-ECG (Lalam et al., 2023) | AUROC, AUPRC | Y | Y | Y |
| (Lai et al., 2023) | AUROC, AUPRC, $F_1$, Specificity, Sensitivity, Acc., PPV | Y | N | N |
| METS (Li et al., 2023) | Acc., PPV, Sensitivity, $F_1$ | N | N | Y |
| MaeFE (Zhang et al., 2023a) | AUROC, Acc., $F_1$ | Y | N | N |
| sCL-ST (Le et al., 2023) | AUROC, AUPRC, Acc., $F_1$, $F_2$, $G_2$ | Y | N | N |
| T-S Reverse (Zhang et al., 2023b) | AUROC, Acc., Sensitivity, Specificity | Y | N | N |
| ST-MEM (Na et al., 2024) | AUROC, $F_1$, Acc. | Y | Y | N |
| ESI (Yu et al., 2024) | AUROC, $F_1$, Acc. | N | Y | Y |
| ETP (Liu et al., 2024a) | AUROC, $F_1$, Acc. | Y | N | Y |
| KED (Tian et al., 2024) | AUROC, AUPRC, Acc., $F_1$, MCC, Sensitivity, Specificity | Y | Y | N |
| MERL (Liu et al., 2024b) | AUROC | N | N | Y |
| D-BETA (Hung et al., 2024) | AUROC | N | Y | N |
| MELP (Wang et al., 2025) | AUROC | N | N | N |
| H-tuning (Zhou et al., 2025) | AUROC, $F_2$, $G_2$, PPV | Y | Y | N |
| HeartLang (Jin et al., 2025) | AUROC | N | N | N |
| ECG-FM (McKeen et al., 2025) | AUROC, AUPRC, AUPRG | Y | N | Y |
| ECGFounder (Li et al., 2025) | AUROC, $F_1$, Acc. | Y | Y | N |
| CREMA (Song et al., 2025) | AUROC, AUPRC | Y | N | Y |

# B. Datasets

The public datasets were obtained from PhysioNet (Goldberger et al., 2000). PhysioNet provides a `wget` command for terminal-based download. The command recursively crawls PhysioNet, so transient failures in fetching directory listings can silently omit files. We encountered this issue while reproducing the data from our original submission on a separate machine: different servers produced different incomplete local copies of PTB-XL, CPSC2018, and CSN. To address this, we provide scripts in our code release that use PhysioNet checksum files to verify completeness and correctness, and iteratively redownloads missing or corrupted files. We have since reseeded our experiments and corrected the reported numbers. The results were reproduced on two machines, and none of the conclusions in our paper were affected.

## B.1. PTB-XL

PTB-XL (Wagner et al., 2020; 2022) is a dataset of 21,837 12-lead 10-second ECG recordings from 18,885 patients. We follow the field's conventional benchmarking protocol outlined in (Strodthoff et al., 2021). The dataset is often analyzed as four subsets. Each subset has a different number of records, as well as number of associated labels. SUPER includes 21,388 ECGs, SUB includes 21,388 ECGs, RHYTHM includes 21,030 ECGs, and FORM includes 8,978 ECGs. For each subset, we detail the prevalence and total number of positive examples of each label in each split in the following tables: SUPER (Table 10), SUB (Table 11), RHYTHM (Table 12), and FORM (Table 13).

*Table 10.* Downstream tasks with their definition in PTB-XL SUPER. For each label, we report the prevalence of the positive class in the whole dataset. We then report the total number of positive examples in the train, validation, and test set after splitting.

| Task | Description | Prevalence (%) | N Train | N Val | N Test |
|------|-------------|----------------|---------|-------|--------|
| CD | Conduction Disturbance | 22.90% | 3907 | 495 | 496 |
| HYP | Hypertrophy | 12.39% | 2119 | 268 | 262 |
| MI | Myocardial Infarction | 25.57% | 4379 | 540 | 550 |
| NORM | Normal ECG | 44.48% | 7596 | 955 | 963 |
| STTC | ST/T Change | 24.48% | 4186 | 528 | 521 |

*Table 11.* Downstream tasks with their definition in PTB-XL SUB. For each label, we report the prevalence of the positive class in the whole dataset. We then report the total number of positive examples in the train, validation, and test set after splitting.

| Task | Description | Prevalence (%) | N Train | N Val | N Test |
|------|-------------|----------------|---------|-------|--------|
| AMI | Anterior myocardial infarction | 14.39% | 2466 | 306 | 306 |
| CLBBB | Complete left bundle branch block | 2.51% | 428 | 54 | 54 |
| CRBBB | Complete right bundle branch block | 2.53% | 432 | 55 | 54 |
| ILBBB | Incomplete left bundle branch block | 0.36% | 62 | 7 | 8 |
| IMI | Inferior myocardial infarction | 15.29% | 2618 | 326 | 327 |
| IRBBB | Incomplete right bundle branch block | 5.23% | 894 | 112 | 112 |
| ISCA | Ischemic in lateral leads | 4.40% | 756 | 92 | 93 |
| ISCI | Ischemic in inferolateral leads | 1.86% | 318 | 39 | 40 |
| ISC_ | Non-specific ischemic | 5.95% | 1019 | 125 | 128 |
| IVCD | Non-specific intraventricular conduction disturbance | 3.68% | 630 | 78 | 79 |
| LAFB/LPFB | Left anterior/posterior fascicular block | 8.40% | 1437 | 181 | 179 |
| LAO/LAE | Left atrial overload/enlargement | 1.99% | 341 | 43 | 42 |
| LMI | Lateral myocardial infarction | 0.94% | 161 | 20 | 20 |
| LVH | Left ventricular hypertrophy | 9.97% | 1708 | 210 | 214 |
| NORM | Normal ECG | 44.48% | 7596 | 955 | 963 |
| NST_ | Non-specific ST changes | 3.59% | 615 | 75 | 77 |
| PMI | Posterior myocardial infarction | 0.08% | 13 | 2 | 2 |
| RAO/RAE | Right atrial overload/enlargement | 0.46% | 79 | 10 | 10 |
| RVH | Right ventricular hypertrophy | 0.59% | 102 | 12 | 12 |
| SEHYP | Septal hypertrophy | 0.14% | 24 | 3 | 2 |
| STTC | ST/T Change | 10.47% | 1792 | 225 | 222 |
| WPW | Wolff-Parkinson-White syndrome | 0.37% | 64 | 7 | 8 |
| _AVB | AV block | 3.85% | 658 | 83 | 82 |

*Table 12.* Downstream tasks with their definition in PTB-XL RHYTHM. For each label, we report the prevalence of the positive class in the whole dataset. We then report the total number of positive examples in the train, validation, and test set after splitting.

| Task | Description | Prevalence (%) | N Train | N Val | N Test |
|------|-------------|---------------|---------|-------|--------|
| AFIB | Atrial fibrillation | 7.20% | 1211 | 151 | 152 |
| AFLT | Atrial flutter | 0.35% | 59 | 7 | 7 |
| BIGU | Bigeminal pattern (unknown origin, SV/Ventricular) | 0.39% | 66 | 8 | 8 |
| PACE | Normal functioning artificial pacemaker | 1.40% | 237 | 29 | 28 |
| PSVT | Paroxysmal supraventricular tachycardia | 0.11% | 19 | 3 | 2 |
| SARRH | Sinus arrhythmia | 3.67% | 618 | 77 | 77 |
| SBRAD | Sinus bradycardia | 3.03% | 509 | 64 | 64 |
| SR | Sinus rhythm | 79.64% | 13404 | 1670 | 1674 |
| STACH | Sinus tachycardia | 3.93% | 661 | 83 | 82 |
| SVARR | Supraventricular arrhythmia | 0.75% | 128 | 15 | 14 |
| SVTAC | Supraventricular tachycardia | 0.13% | 21 | 3 | 3 |
| TRIGU | Trigeminal pattern (unknown origin, SV/Ventricular) | 0.10% | 16 | 2 | 2 |

*Table 13.* Downstream tasks with their definition in PTB-XL FORM. For each label, we report the prevalence of the positive class in the whole dataset. We then report the total number of positive examples in the train, validation, and test set after splitting.

| Task | Description | Prevalence (%) | N Train | N Val | N Test |
|------|-------------|---------------|---------|-------|--------|
| ABQRS | Abnormal QRS | 37.06% | 2683 | 322 | 322 |
| DIG | Digitalis-effect | 2.02% | 145 | 18 | 18 |
| HVOLT | High QRS voltage | 0.69% | 49 | 7 | 6 |
| INVT | Inverted T-waves | 3.27% | 235 | 30 | 29 |
| LNGQT | Long QT-interval | 1.30% | 94 | 12 | 11 |
| LOWT | Low amplitude T-waves | 4.88% | 350 | 44 | 44 |
| LPR | Prolonged PR interval | 3.79% | 272 | 34 | 34 |
| LVOLT | Low QRS voltages in the frontal and horizontal leads | 2.03% | 145 | 19 | 18 |
| NDT | Non-diagnostic T abnormalities | 20.33% | 1461 | 182 | 182 |
| NST_ | Non-specific ST changes | 8.54% | 615 | 75 | 77 |
| NT_ | Non-specific T-wave changes | 4.71% | 340 | 41 | 42 |
| PAC | Atrial premature complex | 4.43% | 318 | 40 | 40 |
| PRC(S) | Premature complex(es) | 0.11% | 8 | 1 | 1 |
| PVC | Ventricular premature complex | 12.73% | 915 | 114 | 114 |
| QWAVE | Q waves present | 6.10% | 438 | 55 | 55 |
| STD_ | Non-specific ST depression | 11.24% | 807 | 101 | 101 |
| STE_ | Non-specific ST elevation | 0.31% | 22 | 3 | 3 |
| TAB_ | T-wave abnormality | 0.39% | 28 | 4 | 3 |
| VCLVH | Voltage criteria for left ventricular hypertrophy | 9.75% | 701 | 87 | 87 |

## B.2. CPSC2018

The China Physiological Signal Challenge 2018 (CPSC2018) (Liu et al., 2018) is a dataset of 6,877 ECGs sampled at 500 Hz. Recording duration varies between 5 and 72 seconds. We exclude recordings shorter than 10 seconds, and for longer recordings, clip them to 10 seconds. We are left with 6,867 ECGs used for downstream evaluation.

The dataset is multi-label with 9 tasks. The prevalence and total number of positive examples for each split is reported in Table 14.

*Table 14.* Downstream tasks with their definition in CPSC2018. For each label, we report the prevalence of the positive class in the whole dataset. We then report the total number of positive examples in the train, validation, and test set after splitting.

| Task | Description | Prevalence (%) | N Train | N Val | N Test |
|------|-------------|----------------|---------|-------|--------|
| AF   | Atrial fibrillation | 17.77% | 845 | 124 | 251 |
| IAVB | 1st degree AV block | 10.50% | 509 | 80 | 132 |
| LBBB | Left bundle branch block | 3.42% | 175 | 23 | 37 |
| NSR  | Sinus rhythm | 13.37% | 653 | 77 | 188 |
| PAC  | Premature atrial contraction | 8.94% | 420 | 60 | 134 |
| PVC  | Premature ventricular contractions | 10.18% | 498 | 67 | 134 |
| RBBB | Right bundle branch block | 27.00% | 1301 | 188 | 365 |
| STD  | ST depression | 12.64% | 589 | 95 | 184 |
| STE  | ST elevation | 3.20% | 154 | 23 | 43 |

## B.3. CSN

The Chapman-Shaoxing-Ningbo (CSN) database (Zheng et al., 2020) is a dataset of 45,152 10-second, 12-lead ECG recordings from 10,646 patients sampled at 500 Hz. CSN is a multi-label dataset with 63 diagnostic labels. We restrict our experiments to 48 labels, excluding 13 that have no positive examples in the dataset (2AVB2, AVNRT, IDC, LBBB, LBBBB, LVQRSCL, LVQRSLL, MI, MIBW, MIFW, MILW, SAAWR, WAVN) and 2 that have fewer than three positive examples (3AVB, ABI). Diagnostic labels are derived from routine clinical interpretations.

We remove ECGs containing a diagnostic code that is not found in the database's code map, resulting in 31,898 recordings used for downstream evaluation.

ECG recordings are randomly split at the record level into training, validation, and test sets with a ratio of 70/10/20. We note that this dataset did not include associated patient ID with each record to enable a patient-level split. The prevalence of the remaining labels, along with the number of positive examples in each split, is reported in Table 15.

*Table 15.* Downstream tasks with their definition in CSN. For each label, we report the prevalence of the positive class in the whole dataset. We then report the total number of positive examples in the train, validation, and test set after splitting.

| Task | Description | Prevalence (%) | N Train | N Val | N Test |
|---|---|---|---|---|---|
| 1AVB | 1 degree atrioventricular block | 2.19% | 476 | 80 | 144 |
| 2AVB | 2 degree atrioventricular block | 0.07% | 14 | 2 | 6 |
| 2AVB1 | 2 degree atrioventricular block(Type one) | 0.05% | 10 | 1 | 5 |
| AF | Atrial Flutter | 10.46% | 2388 | 324 | 624 |
| AFIB | Atrial Fibrillation | 4.43% | 975 | 147 | 291 |
| ALS | Axis left shift | 2.89% | 651 | 88 | 182 |
| APB | Atrial premature beats | 2.14% | 483 | 71 | 130 |
| AQW | Abnormal Q wave | 1.85% | 409 | 56 | 126 |
| ARS | Axis right shift | 1.66% | 363 | 47 | 121 |
| AT | Atrial Tachycardia | 0.54% | 116 | 15 | 41 |
| AVB | Atrioventricular block | 0.55% | 121 | 13 | 42 |
| AVRT | Atrioventricular Reentrant Tachycardia | 0.02% | 4 | 1 | 2 |
| CCR | Counterclockwise rotation | 0.44% | 98 | 15 | 28 |
| CR | Clockwise rotation | 0.24% | 54 | 10 | 11 |
| ERV | Early repolarization of the ventricles | 0.83% | 172 | 29 | 65 |
| FQRS | FQRS Wave | 0.01% | 1 | 1 | 1 |
| IVB | Intraventricular block | 1.35% | 308 | 40 | 81 |
| JEB | Junctional escape beat | 0.08% | 18 | 3 | 3 |
| JPT | Junctional premature beat | 0.02% | 3 | 2 | 2 |
| LFBBB | Left front bundle branch block | 0.66% | 143 | 22 | 44 |
| LVH | Left ventricular hypertrophy | 0.35% | 80 | 11 | 22 |
| LVQRSAL | Lower voltage QRS in all leads | 2.35% | 529 | 75 | 145 |
| MISW | Myocardial infarction in the side wall | 0.18% | 37 | 9 | 10 |
| PRIE | PR interval extension | 0.09% | 22 | 4 | 3 |
| PWC | P wave Change | 0.27% | 62 | 10 | 15 |
| QTIE | QT interval extension | 0.58% | 114 | 18 | 53 |
| RAH | Right atrial hypertrophy | 0.02% | 4 | 1 | 1 |
| RBBB | Right bundle branch block | 1.67% | 373 | 46 | 114 |
| RVH | Right ventricle hypertrophy | 0.09% | 20 | 4 | 5 |
| SA | Sinus Irregularity | 6.56% | 1445 | 230 | 418 |
| SB | Sinus Bradycardia | 40.16% | 8947 | 1267 | 2596 |
| SR | Sinus Rhythm | 22.31% | 4986 | 712 | 1417 |
| ST | Sinus Tachycardia | 16.12% | 3595 | 506 | 1039 |
| STDD | ST drop down | 1.34% | 301 | 40 | 86 |
| STE | ST extension | 1.30% | 277 | 40 | 99 |
| STTC | ST-T Change | 2.75% | 602 | 91 | 183 |
| STTU | ST tilt up | 0.46% | 103 | 17 | 27 |
| SVT | Supraventricular Tachycardia | 1.92% | 429 | 67 | 117 |
| TWC | T wave Change | 14.55% | 3201 | 486 | 953 |
| TWO | T wave opposite | 3.51% | 782 | 96 | 242 |
| UW | U wave | 0.18% | 39 | 7 | 12 |
| VB | Ventricular bigeminy | 0.01% | 1 | 1 | 1 |
| VEB | Ventricular escape beat | 0.07% | 14 | 2 | 5 |
| VET | Ventricular escape trigeminy | 0.02% | 2 | 4 | 1 |
| VFW | Ventricular fusion wave | 0.03% | 6 | 2 | 1 |
| VPB | Ventricular premature beat | 0.74% | 168 | 20 | 49 |
| VPE | Ventricular preexcitation | 0.04% | 8 | 2 | 2 |
| WPW | Wolff Parkinson White Pattern | 0.17% | 39 | 3 | 11 |

## B.4. EchoNext

EchoNext (Poterucha et al., 2025) is a dataset collected at Columbia University Irving Medical Center of 100,000 10-second ECGs with labels derived from contemporaneous echocardiograms. The dataset includes both continuous measurements and binarized labels, the latter of which we focus on in our experiments.

All ECGs were natively sampled at 250 Hz; we linearly interpolate them to 500 Hz. All ECGs in the dataset were z-scored using dataset statistics. The upper 99.9-th and lower 0.1-st percentile of voltages was clipped. The dataset mean and standard deviation were saved, which we use to re-scale all examples back to millivolts.

EchoNext recommends a standard split into four mutually exclusive subsets: train, validation, test, and no_split. The no_split subset is treated as a hold-out set. In our experiments, we use the train, validation, and test sets, which amounts to 82,543 samples. The training set, which includes 72,475 examples, represents 26,218 patients, so includes more than one ECG per patient. However, the validation and test sets, which contain 4,626 and 5,442 examples respectively, only contain the latest ECG per patient.

Table 16 contains a list of all binary labels in the EchoNext dataset. We list the prevalence for each label, as well as the total number of positive examples represented in the train, validation, and test sets. We use standard abbreviations for each task. We note that SHD is a binary label indicating any moderate or severe structural abnormality as defined by meeting the threshold for any of the other labels in this table.

*Table 16.* Downstream tasks with their definition in EchoNext. For each label, we report the prevalence of the positive class in the whole dataset. We then report the total number of positive examples in the train, validation, and test set after splitting.

| Task | Description | Prevalence (%) | N Train | N Val | N Test |
|------|-------------|----------------|---------|-------|--------|
| AR | Moderate or severe aortic regurgitation | 1.22 | 878 | 62 | 66 |
| AS | Moderate or severe aortic stenosis | 4.19 | 2919 | 252 | 286 |
| LVEF $\leq 45$ | Left ventricular ejection fraction is $\leq 45\%$ | 22.76 | 16962 | 866 | 962 |
| LVWT $\geq 13$ | Max of interventricular septum/posterior wall $\geq 1.3$ cm | 23.75 | 17667 | 877 | 1061 |
| MR | Moderate or severe mitral regurgitation | 8.18 | 6137 | 282 | 337 |
| PASP $\geq 45$ | Pulmonary artery systolic pressure is $\geq 45$ mmHg | 18.18 | 13727 | 581 | 699 |
| PEFF | Presence of a moderate or large pericardial effusion | 2.67 | 2079 | 52 | 69 |
| PR | Moderate or severe pulmonary regurgitation | 0.78 | 603 | 21 | 20 |
| RVSD | Moderate or severe right ventricular systolic dysfunction | 12.58 | 9597 | 368 | 419 |
| SHD | Any moderate or severe structural heart disease | 51.20 | 37958 | 1990 | 2318 |
| TR-MAX $\geq 32$ | Maximum tricuspid regurgitation velocity is $\geq 3.2$ m/s | 9.85 | 7492 | 267 | 375 |
| TR | Moderate or severe tricuspid regurgitation | 10.13 | 7707 | 305 | 353 |

### B.5. Hemodynamic Inference

For hemodynamic inference we use a private dataset of 9,226 10-second 12-lead ECGs collected from 5,072 patients at Massachusetts General Hospital (MGH), originally introduced by (Schlesinger et al., 2022). Each ECG is paired with contemporaneous invasive hemodynamic measurements obtained via right heart catheterization, which serve as ground-truth labels.

We consider two binary classification tasks: inferring elevated mean pulmonary capillary wedge pressure (mPCWP) and elevated mean pulmonary artery pressure (mPA). Measurements are binarized with mPCWP $\geq 15$ mmHg and mPA $\geq 20$ mmHg indicating a positive label.

We construct patient-level splits to avoid information leakage across sets. Patients are randomly divided into training (70%), validation (10%), and test (20%) cohorts. The training set may contain multiple ECGs per patient. We only retain one ECG per patient in the validation and test set; if multiple ECGs are present for a given patient, then a single ECG is selected uniformly at random.

After processing, we are left with 6,458 examples in the training set, 507 in the validation set, and 1,015 in the test set. Table 17 summarizes the two downstream hemodynamic inference tasks, including label definitions, overall prevalence, and the number of positive examples in each split.

*Table 17.* Downstream hemodynamic inference tasks. For each binary label, we report the prevalence of the positive class in the full dataset, as well as the number of positive examples in the train, validation, and test sets.

| Task | Description | Prevalence (%) | N Train | N Val | N Test |
|---|---|---|---|---|---|
| mPA | Mean pulmonary arterial pressure $\geq 20$ mmHg measured by right heart catheterization | 68.16 | 4,297 | 393 | 751 |
| mPCWP | Mean pulmonary capillary wedge pressure $\geq 15$ mmHg measured by right heart catheterization | 49.30 | 3,066 | 306 | 561 |

### B.6. Patient Forecasting

For patient forecasting we evaluate on the risk of developing heart failure within 1 year of an ECG (1YR-HF). In particular, we frame this as a binary prediction task with the outcome defined with echocardiographic ground truth as a left ventricular ejection fraction (LVEF) below 40%. We rely on a private longitudinal dataset collected at Massachusetts General Hospital (MGH), originally introduced by (Bergamaschi et al., 2025). The full dataset contains 913,420 10-second 12-lead ECGs from 82,244 patients and is designed to support long-term outcome prediction from ECGs. After filtering examples that have data for the given task, and those with `NaN` or `Inf` values, we are left with 426,081 ECGs from 46,694 patients.

We use patient-level data splits following the original dataset construction, assigning all ECGs from a given patient to the same split to prevent information leakage across sets. Patients are split into training (75%), validation (10%), and test (15%) cohorts, with all ECGs from a given patient assigned to the same split. Table 18 summarizes the patient forecasting task, including the label definition, prevalence, and number of positive examples in each split.

*Table 18.* Patient forecasting task. For the binary outcome, we report the prevalence of the positive class in the full dataset, as well as the number of positive examples in the train, validation, and test sets.

| Task | Description | Prevalence (%) | N Train | N Val | N Test |
|---|---|---|---|---|---|
| 1YR-HF | Development of heart failure within one year, defined as left ventricular ejection fraction $< 40\%$ on an echocardiogram | 29.49 | 93,960 | 12,929 | 18,774 |

## C. Training Details

### C.1. Linear Probing

To evaluate the quality of learned representations, we freeze each encoder then perform linear probing on the embeddings. We train a separate $\ell_2$-regularized logistic regression model for each label. For our implementation we use `scikit-learn`.

We select hyperparameters using a grid search on the validation set. The sweep considers three hyperparameters: `scale`, `inverse_strength`, and `class_weight`. The `scale` hyperparameter controls whether feature standardization is applied. When enabled, each embedding dimension is rescaled to zero mean and unit variance using statistics computed on the training set only. The hyperparameter `inverse_strength` controls the strength of $\ell_2$ regularization in the logistic regression classifier. Smaller values of `inverse_strength` correspond to stronger regularization, while larger values allow the classifier to fit the training data more closely. The `class_weight` hyperparameter determines how class imbalance is handled during training. When set to "balanced", examples are weighted inversely proportional to their class frequencies in the training data, giving greater weight to examples from the minority class. Otherwise, no class weighting is applied, so each training example contributes equally to the loss. We swept over

$$\texttt{scale} \in \{\text{True}, \text{False}\}$$
$$\texttt{inverse\_strength} \in \{0.01, 0.1, 1.0, 10.0\}$$
$$\texttt{class\_weight} \in \{\text{None}, \text{balanced}\}.$$

All logistic regression models are trained to convergence with a maximum of 10,000 optimization iterations. The best-performing configuration is selected based on validation loss and is evaluated once on the held-out test set.

### C.2. CLOCS

**Background.** CLOCS (Kiyasseh et al., 2021) is a popular ECG-specific self-supervised learning approach that constructs contrastive pairs directly from the temporal structure and lead organization of ECG signals. We use the publicly available CLOCS implementation repository and make several modifications, all of which are open-sourced in our code release.

CLOCS defines a family of contrastive objectives, including contrastive multi-segment coding (CMSC), contrastive multi-lead coding (CMLC), and contrastive multi-segment multi-lead coding (CMSMLC). CMSC constructs positive pairs by sampling multiple temporal segments from the same ECG recording, encouraging representations to be invariant to temporal cropping. CMLC instead constructs positive pairs across different leads of the same ECG, encouraging invariance across lead views. CMSMLC combines both objectives by simultaneously contrasting multiple temporal segments and multiple leads from the same ECG. In the original CLOCS paper, CMSC is reported to achieve the strongest average performance across downstream tasks, and we therefore focus on CMSC for comparison.

In the released CLOCS code base, CMSC is implemented only for single-lead ECGs. To support 12-lead ECG pre-training, we extend this objective by treating each lead as a separate channel, consistent with the approach of (Oh et al., 2022).

**Training Formulation** Given a 10-second ECG recording $\mathbf{x} \in \mathbb{R}^{12 \times T}$, where $T$ is the number of samples, we split the recording in half, producing two 5-second segments $\mathbf{x}^{(1)}$ and $\mathbf{x}^{(2)}$. Each segment is encoded using a shared encoder network, producing $\ell_2$-normalized embeddings $\tilde{\mathbf{z}}^{(1)}, \tilde{\mathbf{z}}^{(2)} \in \mathbb{R}^E$, where $E$ is the embedding dimension.

Given a mini-batch of $B$ ECG recordings, we define the cosine similarity between embeddings with temperature $\tau$ as

$$s_{ij} = \frac{(\tilde{\mathbf{z}}_i^{(1)})^\top \tilde{\mathbf{z}}_j^{(2)}}{\tau}.$$

Segments originating from the same ECG form positive pairs, while segments from different ECGs in the batch serve as negative pairs. We then train CMSC by optimizing a symmetric InfoNCE objective:

$$\mathcal{L} = -\frac{1}{2B} \sum_{i=1}^{B} \left[ \log \frac{\exp(s_{ii})}{\sum_{j=1}^{B} \exp(s_{ij})} + \log \frac{\exp(s_{ii})}{\sum_{j=1}^{B} \exp(s_{ji})} \right].$$

**Training** Pre-training is performed on raw 10-second ECG recordings sampled at 500 Hz from the MIMIC-IV dataset, after removing ECGs containing `NaN` or `Inf` values. The resulting dataset is split into training and validation sets using a

90/10 split, corresponding to 719,394 ECGs for training and 80,641 ECGs for validation. The final model checkpoint is selected based on the minimum contrastive loss on the validation set. Models are trained for 50 epochs, corresponding to a comparable number of optimization steps as used in MERL and D-BETA.

We use the same CNN encoder architecture as the original CLOCS implementation and train using the Adam optimizer (Kingma & Ba, 2017). The original CLOCS paper does not report a hyperparameter grid search and instead fixes the learning rate to $10^{-4}$, the temperature to $\tau = 0.1$, and the batch size to 256. We adopt the same batch size and temperature, and perform a grid search over learning rates $\{10^{-3}, 10^{-4}, 10^{-5}\}$ using the validation set. Based on validation loss, we ultimately selected the model trained with a learning rate of $10^{-4}$. We do not apply signal perturbations during pre-training, as the main results in the CLOCS paper are reported without perturbations.

**Evaluation** Because the CMSC encoder is trained on 5-second views, we apply it to downstream 10-second ECGs at the segment level. We split each 10-second ECG recording $\mathbf{x}$ in half, producing two non-overlapping 5-second segments, $\mathbf{x}^{(1)}$ and $\mathbf{x}^{(2)}$. Each 5-second segment is independently encoded, and those embeddings are used to train a logistic regression classifier, producing class probability vectors $\mathbf{p}^{(1)}$ and $\mathbf{p}^{(2)}$. The final prediction is obtained by averaging the two probability vectors:

$$\mathbf{p} = \frac{1}{2}\left(\mathbf{p}^{(1)} + \mathbf{p}^{(2)}\right).$$

# D. Complete Evaluation on PTB-XL

In Appendix D.1 we show the AUROC and AUPRC for every task within each of the four PTB-XL datasets.

In Appendix D.2 we show the macro-AUROC on PTB-XL SUB, RHYTHM, FORM before and after removing tasks with fewer than 10 positive labels in the test set. We do not show a table for SUPER since each task has far more than 10 positive labels in the test set.

## D.1. Performance On Standard Splits

*Table 19.* Performance for every task in PTB-XL SUPER. Each cell reports AUROC (top line) and AUPRC (bottom line) with 95% confidence intervals.

| Task | Random | CLOCS | KED | HeartLang | MERL | D-BETA |
|---|---|---|---|---|---|---|
| CD | 0.843 0.821–0.863 | 0.794 0.767–0.818 | 0.904 0.887–0.919 | 0.802 0.779–0.824 | 0.893 0.876–0.909 | 0.902 0.885–0.917 |
| | / 0.724 0.689–0.755 | / 0.630 0.590–0.666 | / 0.805 0.776–0.831 | / 0.637 0.600–0.673 | / 0.796 0.765–0.822 | / 0.815 0.788–0.841 |
| HYP | 0.847 0.819–0.873 | 0.840 0.811–0.867 | 0.896 0.875–0.915 | 0.782 0.752–0.813 | 0.873 0.848–0.896 | 0.813 0.786–0.839 |
| | / 0.565 0.510–0.620 | / 0.547 0.486–0.601 | / 0.669 0.614–0.714 | / 0.386 0.336–0.441 | / 0.611 0.556–0.661 | / 0.456 0.395–0.510 |
| MI | 0.841 0.821–0.861 | 0.744 0.719–0.767 | 0.885 0.868–0.900 | 0.801 0.780–0.822 | 0.887 0.871–0.902 | 0.905 0.892–0.919 |
| | / 0.655 0.616–0.698 | / 0.563 0.524–0.597 | / 0.764 0.732–0.792 | / 0.585 0.546–0.628 | / 0.777 0.747–0.802 | / 0.815 0.791–0.838 |
| NORM | 0.891 0.877–0.904 | 0.859 0.843–0.873 | 0.932 0.922–0.942 | 0.875 0.860–0.889 | 0.927 0.916–0.938 | 0.930 0.919–0.939 |
| | / 0.839 0.814–0.863 | / 0.789 0.764–0.815 | / 0.901 0.884–0.917 | / 0.822 0.797–0.846 | / 0.885 0.862–0.904 | / 0.895 0.875–0.913 |
| STTC | 0.881 0.864–0.896 | 0.866 0.848–0.882 | 0.927 0.915–0.938 | 0.860 0.844–0.877 | 0.924 0.911–0.936 | 0.916 0.901–0.928 |
| | / 0.701 0.660–0.737 | / 0.688 0.647–0.726 | / 0.808 0.776–0.838 | / 0.637 0.598–0.677 | / 0.799 0.764–0.830 | / 0.786 0.753–0.817 |

*Table 20.* Performance for every task in PTB-XL SUB. Each cell reports AUROC (top line) and AUPRC (bottom line) with 95% confidence intervals.

| Task | Random | CLOCS | KED | HeartLang | MERL | D-BETA |
|---|---|---|---|---|---|---|
| AMI | 0.894 0.874–0.915 | 0.857 0.832–0.883 | 0.951 0.939–0.962 | 0.876 0.857–0.895 | 0.959 0.949–0.969 | 0.956 0.943–0.967 |
| | / 0.646 0.593–0.700 | / 0.590 0.537–0.646 | / 0.808 0.770–0.845 | / 0.548 0.496–0.601 | / 0.841 0.809–0.873 | / 0.834 0.793–0.869 |
| CLBBB | 0.973 0.934–0.999 | 0.984 0.974–0.993 | 0.998 0.997–1.000 | 0.993 0.984–0.998 | 0.999 0.997–1.000 | 0.999 0.997–1.000 |
| | / 0.880 0.784–0.955 | / 0.676 0.551–0.794 | / 0.949 0.906–0.983 | / 0.867 0.767–0.944 | / 0.947 0.889–0.988 | / 0.970 0.937–0.995 |
| CRBBB | 0.989 0.979–0.996 | 0.982 0.970–0.992 | 0.998 0.996–0.999 | 0.964 0.936–0.983 | 0.998 0.997–0.999 | 0.997 0.995–0.999 |
| | / 0.789 0.691–0.875 | / 0.749 0.647–0.843 | / 0.910 0.838–0.966 | / 0.632 0.490–0.753 | / 0.891 0.788–0.979 | / 0.834 0.733–0.928 |
| ILBBB | 0.899 0.733–0.988 | 0.916 0.812–0.979 | 0.913 0.758–0.994 | 0.876 0.769–0.975 | 0.886 0.683–0.987 | 0.935 0.818–0.993 |
| | / 0.197 0.045–0.499 | / 0.064 0.025–0.119 | / 0.206 0.071–0.472 | / 0.066 0.019–0.140 | / 0.094 0.044–0.160 | / 0.211 0.071–0.470 |
| IMI | 0.864 0.840–0.884 | 0.674 0.644–0.702 | 0.881 0.862–0.899 | 0.750 0.721–0.776 | 0.843 0.819–0.864 | 0.897 0.880–0.914 |
| | / 0.556 0.506–0.609 | / 0.299 0.262–0.341 | / 0.597 0.549–0.647 | / 0.353 0.310–0.399 | / 0.538 0.488–0.589 | / 0.691 0.649–0.734 |
| IRBBB | 0.856 0.815–0.891 | 0.810 0.770–0.850 | 0.955 0.935–0.970 | 0.770 0.726–0.813 | 0.971 0.961–0.980 | 0.929 0.905–0.949 |
| | / 0.341 0.262–0.433 | / 0.227 0.173–0.292 | / 0.617 0.530–0.697 | / 0.187 0.137–0.246 | / 0.683 0.602–0.757 | / 0.517 0.431–0.602 |
| ISCA | 0.803 0.762–0.840 | 0.848 0.809–0.884 | 0.927 0.910–0.944 | 0.794 0.753–0.833 | 0.903 0.871–0.931 | 0.923 0.905–0.938 |
| | / 0.158 0.114–0.209 | / 0.224 0.165–0.294 | / 0.350 0.274–0.447 | / 0.124 0.100–0.154 | / 0.331 0.255–0.423 | / 0.275 0.218–0.346 |
| ISCI | 0.854 0.791–0.903 | 0.738 0.663–0.810 | 0.919 0.883–0.949 | 0.767 0.700–0.830 | 0.842 0.768–0.908 | 0.915 0.876–0.946 |
| | / 0.175 0.089–0.292 | / 0.093 0.037–0.184 | / 0.306 0.194–0.442 | / 0.073 0.044–0.121 | / 0.281 0.152–0.419 | / 0.292 0.178–0.443 |
| ISC_ | 0.911 0.879–0.940 | 0.923 0.899–0.943 | 0.966 0.952–0.977 | 0.911 0.893–0.929 | 0.954 0.933–0.971 | 0.940 0.923–0.954 |
| | / 0.538 0.453–0.623 | / 0.507 0.429–0.588 | / 0.703 0.635–0.769 | / 0.411 0.336–0.488 | / 0.674 0.601–0.746 | / 0.539 0.459–0.617 |
| IVCD | 0.696 0.636–0.759 | 0.622 0.564–0.686 | 0.717 0.655–0.782 | 0.698 0.640–0.753 | 0.749 0.685–0.815 | 0.758 0.700–0.815 |
| | / 0.122 0.079–0.185 | / 0.072 0.051–0.105 | / 0.142 0.095–0.206 | / 0.090 0.063–0.124 | / 0.192 0.127–0.273 | / 0.145 0.097–0.210 |
| LAFB/LPFB | 0.941 0.919–0.960 | 0.863 0.833–0.890 | 0.962 0.948–0.974 | 0.878 0.853–0.901 | 0.923 0.903–0.940 | 0.951 0.933–0.968 |
| | / 0.719 0.652–0.776 | / 0.440 0.372–0.511 | / 0.767 0.708–0.824 | / 0.454 0.384–0.525 | / 0.604 0.530–0.674 | / 0.770 0.714–0.821 |
| LAO/LAE | 0.690 0.612–0.759 | 0.648 0.567–0.725 | 0.828 0.771–0.881 | 0.712 0.627–0.797 | 0.818 0.749–0.880 | 0.835 0.778–0.882 |
| | / 0.042 0.029–0.060 | / 0.040 0.027–0.062 | / 0.135 0.070–0.228 | / 0.088 0.041–0.167 | / 0.146 0.083–0.244 | / 0.114 0.063–0.192 |
| LMI | 0.815 0.717–0.903 | 0.657 0.538–0.770 | 0.756 0.657–0.845 | 0.744 0.657–0.825 | 0.741 0.642–0.826 | 0.771 0.703–0.837 |
| | / 0.072 0.033–0.130 | / 0.019 0.012–0.030 | / 0.085 0.021–0.203 | / 0.041 0.017–0.086 | / 0.027 0.016–0.044 | / 0.025 0.017–0.037 |
| LVH | 0.917 0.896–0.935 | 0.900 0.875–0.921 | 0.949 0.935–0.961 | 0.823 0.794–0.853 | 0.929 0.912–0.945 | 0.862 0.837–0.885 |
| | / 0.648 0.593–0.704 | / 0.593 0.528–0.650 | / 0.745 0.695–0.794 | / 0.405 0.343–0.472 | / 0.680 0.623–0.734 | / 0.489 0.426–0.552 |
| NORM | 0.891 0.877–0.903 | 0.859 0.843–0.872 | 0.932 0.921–0.942 | 0.875 0.860–0.890 | 0.926 0.915–0.937 | 0.929 0.919–0.939 |
| | / 0.839 0.813–0.863 | / 0.788 0.763–0.812 | / 0.900 0.883–0.916 | / 0.822 0.798–0.846 | / 0.885 0.862–0.904 | / 0.894 0.875–0.912 |
| NST_ | 0.710 0.649–0.768 | 0.725 0.668–0.780 | 0.835 0.791–0.876 | 0.743 0.693–0.793 | 0.840 0.796–0.880 | 0.838 0.794–0.880 |
| | / 0.136 0.081–0.213 | / 0.142 0.084–0.215 | / 0.190 0.126–0.267 | / 0.111 0.074–0.167 | / 0.193 0.135–0.267 | / 0.180 0.123–0.253 |
| PMI | 0.521 0.173–0.874 | 0.632 0.586–0.679 | 0.932 0.858–0.999 | 0.843 0.679–0.999 | 0.995 0.988–1.000 | 0.797 0.595–0.992 |
| | / 0.003 0.001–0.007 | / 0.002 0.002–0.003 | / 0.159 0.006–0.504 | / 0.110 0.003–0.400 | / 0.241 0.071–0.667 | / 0.031 0.002–0.105 |
| RAO/RAE | 0.866 0.738–0.964 | 0.766 0.556–0.924 | 0.961 0.927–0.988 | 0.919 0.871–0.961 | 0.952 0.902–0.992 | 0.910 0.845–0.968 |
| | / 0.052 0.022–0.104 | / 0.031 0.011–0.075 | / 0.273 0.071–0.541 | / 0.104 0.027–0.287 | / 0.303 0.099–0.586 | / 0.110 0.028–0.287 |
| RVH | 0.862 0.749–0.961 | 0.894 0.725–0.988 | 0.966 0.931–0.990 | 0.903 0.791–0.979 | 0.863 0.683–0.991 | 0.966 0.935–0.986 |
| | / 0.405 0.136–0.691 | / 0.236 0.085–0.456 | / 0.294 0.104–0.541 | / 0.318 0.080–0.598 | / 0.338 0.111–0.594 | / 0.211 0.076–0.396 |
| SEHYP | 0.963 0.942–0.982 | 0.861 0.756–0.956 | 0.993 0.989–0.996 | 0.840 0.701–0.975 | 0.917 0.872–0.958 | 0.889 0.850–0.925 |
| | / 0.025 0.016–0.050 | / 0.009 0.004–0.021 | / 0.104 0.061–0.182 | / 0.013 0.003–0.036 | / 0.011 0.007–0.022 | / 0.008 0.006–0.012 |
| STTC | 0.827 0.801–0.851 | 0.823 0.796–0.849 | 0.893 0.873–0.911 | 0.799 0.772–0.825 | 0.885 0.863–0.905 | 0.862 0.837–0.887 |
| | / 0.319 0.275–0.368 | / 0.319 0.279–0.366 | / 0.507 0.444–0.570 | / 0.295 0.250–0.346 | / 0.480 0.421–0.539 | / 0.429 0.370–0.490 |
| WPW | 0.710 0.490–0.909 | 0.777 0.584–0.924 | 0.968 0.924–0.996 | 0.895 0.753–0.983 | 0.947 0.849–0.999 | 0.840 0.622–0.980 |
| | / 0.120 0.007–0.385 | / 0.031 0.009–0.083 | / 0.460 0.142–0.796 | / 0.261 0.028–0.543 | / 0.621 0.306–0.904 | / 0.439 0.135–0.811 |
| _AVB | 0.743 0.679–0.802 | 0.707 0.649–0.767 | 0.957 0.940–0.971 | 0.852 0.812–0.884 | 0.964 0.949–0.976 | 0.976 0.965–0.984 |
| | / 0.123 0.089–0.174 | / 0.130 0.083–0.196 | / 0.564 0.462–0.664 | / 0.223 0.157–0.303 | / 0.575 0.479–0.684 | / 0.625 0.517–0.734 |

*Table 21.* Performance for every task in PTB-XL RHYTHM. Each cell reports AUROC (top line) and AUPRC (bottom line) with 95% confidence intervals.

| Task | Random | CLOCS | KED | HeartLang | MERL | D-BETA |
|------|--------|-------|-----|-----------|------|--------|
| AFIB | 0.858 0.827–0.888 / 0.384 0.318–0.463 | 0.836 0.803–0.867 / 0.337 0.277–0.410 | 0.984 0.967–0.995 / 0.936 0.903–0.964 | 0.928 0.908–0.947 / 0.577 0.500–0.651 | 0.981 0.965–0.992 / 0.891 0.839–0.937 | 0.984 0.967–0.997 / 0.955 0.924–0.979 |
| AFLT | 0.948 0.897–0.991 / 0.231 0.046–0.531 | 0.911 0.748–0.997 / 0.302 0.090–0.632 | 0.966 0.906–0.999 / 0.508 0.178–0.824 | 0.795 0.527–0.999 / 0.451 0.117–0.801 | 0.969 0.914–0.999 / 0.611 0.277–0.901 | 0.950 0.859–1.000 / 0.582 0.241–0.883 |
| BIGU | 0.636 0.405–0.871 / 0.102 0.011–0.286 | 0.703 0.498–0.896 / 0.147 0.006–0.408 | 0.955 0.893–0.996 / 0.445 0.139–0.751 | 0.884 0.790–0.969 / 0.106 0.020–0.309 | 0.744 0.541–0.926 / 0.148 0.015–0.427 | 0.975 0.960–0.989 / 0.350 0.071–0.669 |
| PACE | 0.905 0.830–0.967 / 0.589 0.410–0.759 | 0.889 0.801–0.957 / 0.428 0.252–0.597 | 0.971 0.937–0.994 / 0.733 0.579–0.864 | 0.905 0.810–0.979 / 0.551 0.371–0.721 | 0.962 0.900–0.997 / 0.796 0.657–0.916 | 0.993 0.985–0.999 / 0.887 0.776–0.973 |
| PSVT | 0.921 0.834–0.998 / 0.096 0.006–0.333 | 0.998 0.994–1.000 / 0.613 0.143–1.000 | 0.999 0.996–1.000 / 0.664 0.200–1.000 | 0.998 0.995–1.000 / 0.626 0.166–1.000 | 0.998 0.995–1.000 / 0.477 0.167–1.000 | 0.999 0.996–1.000 / 0.664 0.200–1.000 |
| SARRH | 0.644 0.581–0.708 / 0.079 0.053–0.114 | 0.611 0.552–0.676 / 0.063 0.046–0.092 | 0.873 0.837–0.907 / 0.233 0.170–0.309 | 0.625 0.557–0.683 / 0.059 0.045–0.079 | 0.690 0.631–0.743 / 0.099 0.065–0.146 | 0.950 0.917–0.970 / 0.515 0.414–0.618 |
| SBRAD | 0.791 0.733–0.850 / 0.173 0.105–0.263 | 0.891 0.846–0.932 / 0.430 0.320–0.559 | 0.968 0.953–0.980 / 0.677 0.571–0.775 | 0.938 0.907–0.962 / 0.487 0.373–0.595 | 0.962 0.945–0.976 / 0.565 0.459–0.675 | 0.933 0.885–0.967 / 0.621 0.519–0.726 |
| SR | 0.690 0.661–0.718 / 0.872 0.856–0.888 | 0.784 0.755–0.810 / 0.915 0.900–0.929 | 0.933 0.919–0.947 / 0.979 0.973–0.984 | 0.841 0.817–0.863 / 0.941 0.928–0.952 | 0.890 0.871–0.909 / 0.962 0.953–0.970 | 0.949 0.936–0.962 / 0.975 0.963–0.986 |
| STACH | 0.864 0.825–0.899 / 0.224 0.168–0.297 | 0.975 0.967–0.983 / 0.594 0.490–0.697 | 0.992 0.980–0.999 / 0.945 0.899–0.982 | 0.987 0.976–0.995 / 0.851 0.775–0.912 | 0.993 0.989–0.996 / 0.860 0.790–0.922 | 0.989 0.968–0.999 / 0.922 0.859–0.971 |
| SVARR | 0.862 0.779–0.926 / 0.040 0.022–0.066 | 0.602 0.507–0.692 / 0.009 0.007–0.012 | 0.924 0.869–0.971 / 0.344 0.124–0.580 | 0.815 0.753–0.875 / 0.023 0.015–0.037 | 0.934 0.865–0.980 / 0.425 0.193–0.668 | 0.910 0.811–0.981 / 0.332 0.127–0.579 |
| SVTAC | 0.819 0.643–0.999 / 0.182 0.003–0.669 | 0.962 0.903–0.999 / 0.211 0.015–0.707 | 0.996 0.991–0.999 / 0.251 0.111–0.533 | 0.966 0.908–0.999 / 0.234 0.015–0.744 | 0.993 0.987–0.998 / 0.213 0.078–0.600 | 0.999 0.997–1.000 / 0.474 0.224–1.000 |
| TRIGU | 0.505 0.091–0.865 / 0.003 0.001–0.007 | 0.559 0.156–0.905 / 0.004 0.001–0.010 | 0.942 0.891–0.994 / 0.043 0.009–0.143 | 0.562 0.503–0.623 / 0.002 0.002–0.003 | 0.726 0.461–0.950 / 0.008 0.002–0.019 | 0.989 0.981–0.995 / 0.075 0.045–0.154 |

*Table 22.* Performance for every task in PTB-XL FORM. Each cell reports AUROC (top line) and AUPRC (bottom line) with 95% confidence intervals.

| Task | Random | CLOCS | KED | HeartLang | MERL | D-BETA |
|------|--------|-------|-----|-----------|------|--------|
| ABQRS | 0.788 0.757–0.819 / 0.687 0.639–0.736 | 0.684 0.649–0.717 / 0.538 0.491–0.583 | 0.829 0.798–0.858 / 0.741 0.697–0.785 | 0.731 0.696–0.765 / 0.615 0.568–0.661 | 0.811 0.780–0.839 / 0.720 0.675–0.766 | 0.785 0.757–0.814 / 0.648 0.602–0.698 |
| DIG | 0.755 0.643–0.852 / 0.070 0.039–0.119 | 0.628 0.491–0.756 / 0.045 0.023–0.087 | 0.907 0.861–0.949 / 0.326 0.151–0.534 | 0.714 0.583–0.816 / 0.061 0.032–0.126 | 0.878 0.819–0.929 / 0.149 0.076–0.269 | 0.882 0.806–0.943 / 0.239 0.111–0.431 |
| HVOLT | 0.907 0.839–0.968 / 0.078 0.030–0.184 | 0.829 0.742–0.911 / 0.046 0.017–0.131 | 0.943 0.916–0.967 / 0.080 0.049–0.143 | 0.808 0.622–0.926 / 0.035 0.015–0.062 | 0.936 0.888–0.973 / 0.089 0.043–0.169 | 0.907 0.847–0.948 / 0.052 0.030–0.083 |
| INVT | 0.864 0.819–0.908 / 0.161 0.104–0.250 | 0.864 0.811–0.910 / 0.161 0.103–0.240 | 0.899 0.854–0.940 / 0.421 0.258–0.595 | 0.765 0.679–0.845 / 0.145 0.078–0.244 | 0.910 0.872–0.945 / 0.317 0.183–0.471 | 0.915 0.880–0.948 / 0.353 0.213–0.519 |
| LNGQT | 0.679 0.495–0.847 / 0.046 0.018–0.105 | 0.690 0.535–0.818 / 0.037 0.017–0.082 | 0.804 0.638–0.941 / 0.218 0.051–0.494 | 0.701 0.544–0.858 / 0.058 0.020–0.151 | 0.850 0.716–0.972 / 0.348 0.125–0.620 | 0.898 0.818–0.965 / 0.229 0.067–0.455 |
| LOWT | 0.745 0.666–0.823 / 0.145 0.100–0.202 | 0.726 0.654–0.794 / 0.112 0.081–0.155 | 0.831 0.775–0.881 / 0.240 0.157–0.344 | 0.687 0.616–0.757 / 0.122 0.075–0.191 | 0.833 0.789–0.874 / 0.214 0.139–0.320 | 0.843 0.795–0.885 / 0.186 0.136–0.260 |
| LPR | 0.610 0.510–0.708 / 0.072 0.045–0.123 | 0.588 0.478–0.692 / 0.063 0.042–0.097 | 0.934 0.891–0.967 / 0.436 0.300–0.589 | 0.773 0.701–0.841 / 0.198 0.099–0.322 | 0.958 0.937–0.975 / 0.504 0.344–0.665 | 0.964 0.947–0.978 / 0.446 0.327–0.598 |
| LVOLT | 0.937 0.907–0.960 / 0.216 0.124–0.371 | 0.883 0.830–0.928 / 0.124 0.070–0.230 | 0.940 0.908–0.966 / 0.249 0.138–0.424 | 0.771 0.621–0.889 / 0.212 0.066–0.380 | 0.930 0.893–0.959 / 0.220 0.119–0.393 | 0.809 0.725–0.876 / 0.080 0.044–0.143 |
| NDT | 0.810 0.777–0.842 / 0.507 0.446–0.577 | 0.800 0.769–0.831 / 0.455 0.399–0.513 | 0.896 0.873–0.918 / 0.666 0.605–0.731 | 0.761 0.725–0.793 / 0.432 0.374–0.486 | 0.891 0.868–0.913 / 0.693 0.632–0.753 | 0.866 0.839–0.892 / 0.612 0.549–0.682 |
| NST_ | 0.651 0.587–0.712 / 0.189 0.125–0.270 | 0.644 0.575–0.712 / 0.205 0.139–0.292 | 0.740 0.679–0.799 / 0.252 0.184–0.338 | 0.639 0.578–0.698 / 0.175 0.119–0.247 | 0.766 0.708–0.816 / 0.280 0.205–0.363 | 0.758 0.703–0.810 / 0.259 0.192–0.352 |
| NT_ | 0.755 0.690–0.818 / 0.138 0.089–0.205 | 0.671 0.600–0.740 / 0.087 0.063–0.128 | 0.806 0.750–0.858 / 0.208 0.125–0.327 | 0.693 0.618–0.774 / 0.099 0.071–0.137 | 0.878 0.843–0.908 / 0.223 0.158–0.320 | 0.828 0.764–0.882 / 0.215 0.141–0.305 |
| PAC | 0.614 0.518–0.709 / 0.090 0.056–0.154 | 0.628 0.551–0.705 / 0.080 0.055–0.126 | 0.934 0.894–0.964 / 0.566 0.432–0.704 | 0.684 0.596–0.760 / 0.101 0.068–0.148 | 0.720 0.642–0.796 / 0.147 0.087–0.244 | 0.985 0.975–0.992 / 0.687 0.547–0.825 |
| PRC(S) | 0.923 0.907–0.941 / 0.015 0.012–0.019 | 0.850 0.826–0.874 / 0.008 0.006–0.009 | 0.967 0.956–0.977 / 0.034 0.025–0.048 | 0.867 0.845–0.889 / 0.009 0.007–0.010 | 0.761 0.732–0.790 / 0.005 0.004–0.005 | 0.970 0.959–0.981 / 0.038 0.027–0.056 |
| PVC | 0.837 0.794–0.879 / 0.535 0.443–0.628 | 0.797 0.746–0.844 / 0.452 0.364–0.544 | 0.963 0.949–0.975 / 0.809 0.738–0.870 | 0.938 0.915–0.957 / 0.759 0.688–0.826 | 0.910 0.881–0.937 / 0.665 0.585–0.745 | 0.995 0.991–0.998 / 0.962 0.937–0.983 |
| QWAVE | 0.664 0.587–0.735 / 0.156 0.099–0.231 | 0.574 0.495–0.650 / 0.113 0.072–0.179 | 0.736 0.667–0.803 / 0.191 0.134–0.269 | 0.636 0.562–0.711 / 0.140 0.088–0.217 | 0.712 0.658–0.764 / 0.124 0.093–0.171 | 0.766 0.706–0.823 / 0.193 0.131–0.276 |
| STD_ | 0.750 0.701–0.796 / 0.272 0.218–0.337 | 0.659 0.601–0.715 / 0.235 0.180–0.296 | 0.811 0.769–0.847 / 0.360 0.289–0.440 | 0.668 0.612–0.726 / 0.239 0.184–0.310 | 0.809 0.769–0.849 / 0.392 0.317–0.476 | 0.753 0.706–0.798 / 0.285 0.227–0.352 |
| STE_ | 0.642 0.406–0.783 / 0.008 0.005–0.014 | 0.562 0.100–0.908 / 0.011 0.004–0.036 | 0.585 0.295–0.936 / 0.014 0.004–0.051 | 0.612 0.202–0.930 / 0.014 0.004–0.047 | 0.921 0.840–0.993 / 0.112 0.018–0.429 | 0.710 0.511–0.949 / 0.018 0.006–0.062 |
| TAB_ | 0.562 0.390–0.838 / 0.008 0.004–0.021 | 0.696 0.591–0.797 / 0.009 0.006–0.016 | 0.741 0.515–0.976 / 0.034 0.006–0.130 | 0.273 0.135–0.498 / 0.004 0.003–0.007 | 0.868 0.797–0.974 / 0.038 0.013–0.120 | 0.662 0.235–0.954 / 0.021 0.004–0.070 |
| VCLVH | 0.866 0.827–0.902 / 0.437 0.346–0.535 | 0.809 0.758–0.855 / 0.326 0.257–0.406 | 0.905 0.872–0.931 / 0.505 0.412–0.610 | 0.707 0.651–0.763 / 0.211 0.162–0.272 | 0.857 0.816–0.895 / 0.416 0.328–0.516 | 0.766 0.718–0.816 / 0.299 0.229–0.389 |

## D.2. PTB-XL Results When Labels with Minimal Examples Are Removed

Because macro-AUROC weights the performance on each label equally, omitting a few classes with extremely small test support can disproportionately affect whichever method happened to perform the worst on those labels. In PTB-XL SUB (Table 23) and PTB-XL RHYTHM (Table 24), Random experiences the largest swing in macro-AUROC after evaluating on the cleaned data. We don't think this is an intrinsic property of the Random encoder. For example, in PTB-XL FORM, all other methods changed more than Random (Table 25).

*Table 23.* ORIG uses the standard PTB-XL SUB test set; CLEAN excludes labels with fewer than 10 positive test examples. Values are macro-AUROC with 95% confidence intervals.

| Method | ORIG | CLEAN | ΔMacro-AUROC |
|---|---|---|---|
| RANDOM | 0.834 0.808–0.859 | 0.847 0.834–0.860 | +0.013 |
| CLOCS | 0.803 0.784–0.819 | 0.803 0.787–0.820 | +0.000 |
| KED | 0.920 0.909–0.929 | 0.913 0.905–0.921 | -0.007 |
| HEARTLANG | 0.836 0.819–0.853 | 0.830 0.819–0.841 | -0.006 |
| MERL | 0.905 0.890–0.918 | 0.897 0.883–0.909 | -0.007 |
| D-BETA | 0.899 0.882–0.914 | 0.906 0.899–0.914 | +0.007 |

*Table 24.* ORIG uses the standard PTB-XL RHYTHM test set; CLEAN excludes labels with fewer than 10 positive test examples. Values are macro-AUROC with 95% confidence intervals.

| Method | ORIG | CLEAN | ΔMacro-AUROC |
|---|---|---|---|
| RANDOM | 0.787 0.737–0.833 | 0.802 0.781–0.823 | +0.015 |
| CLOCS | 0.810 0.762–0.854 | 0.798 0.775–0.819 | -0.013 |
| KED | 0.959 0.948–0.969 | 0.949 0.938–0.959 | -0.009 |
| HEARTLANG | 0.854 0.827–0.879 | 0.861 0.841–0.879 | +0.008 |
| MERL | 0.903 0.870–0.933 | 0.914 0.898–0.927 | +0.011 |
| D-BETA | 0.968 0.956–0.978 | 0.958 0.941–0.971 | -0.010 |

*Table 25.* ORIG uses the standard PTB-XL FORM test set; CLEAN excludes labels with fewer than 10 positive test examples. Values are macro-AUROC with 95% confidence intervals.

| Method | ORIG | CLEAN | ΔMacro-AUROC |
|---|---|---|---|
| RANDOM | 0.756 0.734–0.780 | 0.754 0.733–0.774 | -0.002 |
| CLOCS | 0.715 0.687–0.741 | 0.710 0.690–0.731 | -0.005 |
| KED | 0.851 0.827–0.875 | 0.862 0.846–0.876 | +0.011 |
| HEARTLANG | 0.707 0.679–0.736 | 0.723 0.699–0.745 | +0.016 |
| MERL | 0.853 0.839–0.866 | 0.847 0.833–0.860 | -0.005 |
| D-BETA | 0.845 0.821–0.868 | 0.855 0.841–0.868 | +0.009 |

# E. Complete Evaluation on CPSC2018

*Table 26.* Performance on CPSC2018 sub-tasks. Each cell reports AUROC (top line) and AUPRC (bottom line), each with a 95% confidence interval.

| Task | Random | CLOCS | KED | HeartLang | MERL | D-BETA |
|------|--------|-------|-----|-----------|------|--------|
| AF | 0.873 $_{0.848-0.897}$ / 0.660 $_{0.605-0.714}$ | 0.837 $_{0.807-0.862}$ / 0.617 $_{0.565-0.669}$ | 0.987 $_{0.979-0.993}$ / 0.957 $_{0.938-0.973}$ | 0.914 $_{0.897-0.930}$ / 0.700 $_{0.650-0.750}$ | 0.979 $_{0.967-0.989}$ / 0.947 $_{0.923-0.968}$ | 0.995 $_{0.993-0.998}$ / 0.983 $_{0.974-0.991}$ |
| IAVB | 0.774 $_{0.732-0.814}$ / 0.296 $_{0.235-0.366}$ | 0.706 $_{0.660-0.751}$ / 0.211 $_{0.167-0.265}$ | 0.987 $_{0.980-0.993}$ / 0.922 $_{0.889-0.950}$ | 0.818 $_{0.780-0.859}$ / 0.437 $_{0.356-0.521}$ | 0.984 $_{0.974-0.992}$ / 0.916 $_{0.880-0.946}$ | 0.993 $_{0.986-0.997}$ / 0.954 $_{0.927-0.975}$ |
| LBBB | 0.946 $_{0.868-0.999}$ / 0.839 $_{0.693-0.965}$ | 0.976 $_{0.960-0.991}$ / 0.758 $_{0.638-0.877}$ | 0.996 $_{0.990-0.999}$ / 0.931 $_{0.861-0.983}$ | 0.962 $_{0.914-0.997}$ / 0.855 $_{0.751-0.944}$ | 0.987 $_{0.967-0.998}$ / 0.898 $_{0.820-0.964}$ | 0.998 $_{0.996-1.000}$ / 0.957 $_{0.910-0.991}$ |
| NSR | 0.893 $_{0.871-0.913}$ / 0.593 $_{0.532-0.658}$ | 0.846 $_{0.820-0.872}$ / 0.461 $_{0.400-0.523}$ | 0.958 $_{0.946-0.969}$ / 0.804 $_{0.752-0.850}$ | 0.893 $_{0.870-0.916}$ / 0.585 $_{0.516-0.658}$ | 0.951 $_{0.938-0.963}$ / 0.760 $_{0.699-0.820}$ | 0.955 $_{0.943-0.965}$ / 0.762 $_{0.702-0.816}$ |
| PAC | 0.706 $_{0.659-0.749}$ / 0.196 $_{0.160-0.240}$ | 0.597 $_{0.554-0.641}$ / 0.121 $_{0.106-0.141}$ | 0.861 $_{0.828-0.892}$ / 0.470 $_{0.390-0.556}$ | 0.659 $_{0.610-0.709}$ / 0.178 $_{0.144-0.222}$ | 0.779 $_{0.734-0.821}$ / 0.334 $_{0.263-0.407}$ | 0.927 $_{0.898-0.952}$ / 0.703 $_{0.622-0.784}$ |
| PVC | 0.748 $_{0.700-0.796}$ / 0.370 $_{0.296-0.444}$ | 0.699 $_{0.649-0.745}$ / 0.220 $_{0.170-0.280}$ | 0.881 $_{0.849-0.910}$ / 0.603 $_{0.528-0.674}$ | 0.840 $_{0.802-0.874}$ / 0.468 $_{0.390-0.552}$ | 0.815 $_{0.778-0.852}$ / 0.460 $_{0.379-0.545}$ | 0.910 $_{0.876-0.939}$ / 0.781 $_{0.717-0.840}$ |
| RBBB | 0.955 $_{0.943-0.966}$ / 0.905 $_{0.877-0.929}$ | 0.927 $_{0.911-0.942}$ / 0.833 $_{0.791-0.869}$ | 0.981 $_{0.974-0.987}$ / 0.949 $_{0.931-0.966}$ | 0.889 $_{0.869-0.907}$ / 0.765 $_{0.724-0.804}$ | 0.983 $_{0.977-0.988}$ / 0.951 $_{0.930-0.969}$ | 0.976 $_{0.967-0.983}$ / 0.937 $_{0.906-0.961}$ |
| STD | 0.875 $_{0.844-0.899}$ / 0.571 $_{0.500-0.640}$ | 0.820 $_{0.788-0.848}$ / 0.435 $_{0.369-0.506}$ | 0.958 $_{0.941-0.972}$ / 0.818 $_{0.765-0.866}$ | 0.826 $_{0.792-0.859}$ / 0.474 $_{0.407-0.538}$ | 0.957 $_{0.940-0.971}$ / 0.835 $_{0.782-0.881}$ | 0.949 $_{0.931-0.966}$ / 0.822 $_{0.766-0.870}$ |
| STE | 0.880 $_{0.811-0.937}$ / 0.414 $_{0.273-0.559}$ | 0.814 $_{0.734-0.887}$ / 0.342 $_{0.202-0.483}$ | 0.955 $_{0.916-0.982}$ / 0.607 $_{0.473-0.732}$ | 0.836 $_{0.773-0.892}$ / 0.263 $_{0.159-0.404}$ | 0.906 $_{0.831-0.963}$ / 0.591 $_{0.458-0.719}$ | 0.923 $_{0.873-0.965}$ / 0.523 $_{0.388-0.662}$ |

# F. Complete Evaluation on CSN

*Table 27.* Performance on CSN sub-tasks whose labels begin with letter A through M. Each cell reports AUROC (top line) and AUPRC (bottom line), each with a 95% confidence interval.

| Task | Random | CLOCS | KED | HeartLang | MERL | D-BETA |
|---|---|---|---|---|---|---|
| 1AVB | 0.773 0.735–0.811 / 0.093 0.071–0.122 | 0.705 0.663–0.746 / 0.057 0.042–0.081 | 0.975 0.953–0.991 / 0.757 0.688–0.817 | 0.865 0.832–0.896 / 0.228 0.173–0.291 | 0.981 0.968–0.990 / 0.719 0.647–0.789 | 0.990 0.986–0.994 / 0.803 0.742–0.853 |
| 2AVB | 0.470 0.266–0.728 / 0.007 0.001–0.033 | 0.936 0.880–0.990 / 0.217 0.011–0.553 | 0.981 0.951–0.997 / 0.103 0.040–0.187 | 0.880 0.718–0.968 / 0.011 0.004–0.020 | 0.867 0.643–0.988 / 0.034 0.008–0.101 | 0.983 0.961–0.995 / 0.066 0.027–0.125 |
| 2AVB1 | 0.593 0.422–0.795 / 0.067 0.001–0.268 | 0.796 0.651–0.931 / 0.015 0.002–0.050 | 0.940 0.897–0.983 / 0.019 0.005–0.055 | 0.575 0.291–0.834 / 0.057 0.001–0.258 | 0.792 0.550–0.954 / 0.008 0.002–0.024 | 0.978 0.952–0.992 / 0.039 0.016–0.067 |
| AF | 0.867 0.852–0.881 / 0.410 0.376–0.444 | 0.844 0.829–0.859 / 0.340 0.309–0.371 | 0.973 0.969–0.976 / 0.712 0.674–0.748 | 0.911 0.902–0.921 / 0.475 0.440–0.510 | 0.975 0.971–0.979 / 0.733 0.697–0.772 | 0.977 0.974–0.981 / 0.757 0.723–0.790 |
| AFIB | 0.856 0.834–0.877 / 0.243 0.207–0.280 | 0.856 0.835–0.874 / 0.221 0.186–0.261 | 0.966 0.961–0.971 / 0.510 0.460–0.566 | 0.909 0.897–0.922 / 0.282 0.246–0.322 | 0.962 0.957–0.966 / 0.473 0.423–0.526 | 0.969 0.964–0.973 / 0.527 0.478–0.581 |
| ALS | 0.974 0.964–0.981 / 0.541 0.472–0.612 | 0.856 0.828–0.882 / 0.180 0.143–0.224 | 0.981 0.975–0.986 / 0.597 0.523–0.668 | 0.893 0.869–0.914 / 0.248 0.199–0.302 | 0.938 0.922–0.952 / 0.398 0.331–0.469 | 0.984 0.979–0.988 / 0.673 0.607–0.739 |
| APB | 0.708 0.663–0.755 / 0.050 0.039–0.066 | 0.682 0.639–0.722 / 0.053 0.036–0.080 | 0.941 0.918–0.960 / 0.464 0.377–0.552 | 0.735 0.690–0.778 / 0.053 0.042–0.066 | 0.799 0.760–0.837 / 0.097 0.071–0.134 | 0.987 0.973–0.994 / 0.724 0.644–0.804 |
| AQW | 0.902 0.871–0.931 / 0.243 0.187–0.312 | 0.749 0.698–0.796 / 0.168 0.106–0.237 | 0.938 0.918–0.957 / 0.433 0.350–0.517 | 0.802 0.758–0.841 / 0.105 0.076–0.143 | 0.922 0.890–0.953 / 0.505 0.414–0.592 | 0.968 0.951–0.982 / 0.571 0.475–0.662 |
| ARS | 0.954 0.932–0.972 / 0.401 0.327–0.478 | 0.896 0.871–0.918 / 0.156 0.118–0.206 | 0.971 0.964–0.978 / 0.414 0.336–0.497 | 0.955 0.941–0.969 / 0.366 0.289–0.449 | 0.939 0.915–0.959 / 0.316 0.246–0.396 | 0.910 0.884–0.935 / 0.326 0.246–0.409 |
| AT | 0.742 0.657–0.822 / 0.041 0.016–0.091 | 0.770 0.698–0.837 / 0.032 0.017–0.055 | 0.933 0.899–0.960 / 0.194 0.103–0.316 | 0.804 0.735–0.863 / 0.040 0.020–0.079 | 0.880 0.829–0.921 / 0.061 0.035–0.104 | 0.978 0.967–0.987 / 0.381 0.241–0.537 |
| AVB | 0.628 0.522–0.735 / 0.025 0.013–0.041 | 0.857 0.805–0.902 / 0.036 0.025–0.052 | 0.926 0.905–0.945 / 0.083 0.048–0.135 | 0.851 0.802–0.897 / 0.062 0.031–0.114 | 0.939 0.905–0.968 / 0.189 0.112–0.298 | 0.972 0.960–0.981 / 0.204 0.129–0.302 |
| AVRT | 0.494 0.004–0.987 / 0.008 0.000–0.024 | 0.963 0.926–0.996 / 0.025 0.004–0.069 | 0.928 0.887–0.967 / 0.005 0.003–0.009 | 0.554 0.302–0.803 / 0.001 0.000–0.002 | 0.954 0.929–0.978 / 0.007 0.004–0.014 | 0.976 0.964–0.987 / 0.013 0.009–0.024 |
| CCR | 0.899 0.851–0.939 / 0.060 0.028–0.119 | 0.919 0.882–0.949 / 0.103 0.027–0.210 | 0.929 0.899–0.956 / 0.138 0.049–0.272 | 0.783 0.697–0.862 / 0.034 0.013–0.077 | 0.872 0.810–0.925 / 0.085 0.025–0.183 | 0.831 0.767–0.891 / 0.044 0.015–0.112 |
| CR | 0.938 0.882–0.983 / 0.118 0.034–0.262 | 0.910 0.765–0.992 / 0.138 0.044–0.310 | 0.945 0.869–0.995 / 0.250 0.076–0.500 | 0.808 0.578–0.960 / 0.039 0.009–0.132 | 0.881 0.768–0.970 / 0.244 0.030–0.503 | 0.933 0.842–0.989 / 0.155 0.031–0.347 |
| ERV | 0.940 0.912–0.963 / 0.245 0.167–0.338 | 0.917 0.886–0.944 / 0.209 0.132–0.304 | 0.973 0.965–0.981 / 0.279 0.207–0.379 | 0.892 0.852–0.926 / 0.138 0.084–0.217 | 0.973 0.964–0.982 / 0.324 0.229–0.436 | 0.952 0.936–0.967 / 0.230 0.150–0.323 |
| FQRS | 0.179 0.169–0.188 / 0.000 0.000–0.000 | 0.186 0.177–0.196 / 0.000 0.000–0.000 | 0.883 0.875–0.891 / 0.001 0.001–0.001 | 0.025 0.021–0.029 / 0.000 0.000–0.000 | 0.231 0.220–0.241 / 0.000 0.000–0.000 | 0.985 0.982–0.988 / 0.011 0.009–0.013 |
| IVB | 0.773 0.710–0.830 / 0.070 0.043–0.111 | 0.807 0.761–0.853 / 0.047 0.036–0.061 | 0.941 0.923–0.956 / 0.177 0.130–0.238 | 0.865 0.827–0.898 / 0.105 0.062–0.159 | 0.915 0.885–0.939 / 0.133 0.095–0.183 | 0.976 0.962–0.986 / 0.468 0.360–0.587 |
| JEB | 0.700 0.415–0.905 / 0.002 0.001–0.005 | 0.552 0.148–0.895 / 0.001 0.000–0.004 | 0.989 0.972–1.000 / 0.280 0.017–0.750 | 0.736 0.633–0.828 / 0.001 0.001–0.002 | 0.880 0.659–0.999 / 0.098 0.001–0.350 | 0.983 0.964–0.996 / 0.044 0.013–0.107 |
| JPT | 0.156 0.093–0.223 / 0.000 0.000–0.000 | 0.303 0.078–0.524 / 0.000 0.000–0.001 | 0.562 0.378–0.753 / 0.001 0.001–0.001 | 0.312 0.193–0.436 / 0.000 0.000–0.001 | 0.058 0.040–0.077 / 0.000 0.000–0.000 | 0.895 0.886–0.903 / 0.003 0.002–0.003 |
| LFBBB | 0.991 0.980–0.998 / 0.652 0.518–0.800 | 0.910 0.844–0.966 / 0.362 0.236–0.506 | 0.990 0.979–0.997 / 0.683 0.554–0.797 | 0.953 0.916–0.983 / 0.487 0.335–0.649 | 0.980 0.960–0.993 / 0.628 0.497–0.757 | 0.993 0.988–0.998 / 0.705 0.571–0.825 |
| LVH | 0.888 0.757–0.983 / 0.157 0.072–0.272 | 0.987 0.979–0.993 / 0.244 0.127–0.408 | 0.995 0.993–0.998 / 0.402 0.269–0.583 | 0.859 0.765–0.930 / 0.059 0.021–0.126 | 0.979 0.964–0.992 / 0.216 0.125–0.344 | 0.968 0.949–0.984 / 0.123 0.064–0.214 |
| LVQRSAL | 0.908 0.880–0.935 / 0.320 0.253–0.396 | 0.856 0.822–0.886 / 0.210 0.154–0.271 | 0.936 0.916–0.954 / 0.397 0.316–0.479 | 0.765 0.723–0.805 / 0.104 0.075–0.141 | 0.922 0.900–0.942 / 0.331 0.259–0.408 | 0.862 0.831–0.888 / 0.196 0.141–0.260 |
| MISW | 0.854 0.665–0.969 / 0.046 0.010–0.131 | 0.782 0.590–0.943 / 0.055 0.009–0.162 | 0.944 0.906–0.974 / 0.076 0.014–0.222 | 0.792 0.660–0.908 / 0.037 0.004–0.157 | 0.916 0.844–0.974 / 0.049 0.014–0.121 | 0.965 0.933–0.990 / 0.090 0.028–0.209 |

*Table 28.* Performance on CSN sub-tasks whose labels begin with letter P through Z. Each cell reports AUROC (top line) and AUPRC (bottom line), each with a 95% confidence interval.

| Task | Random | CLOCS | KED | HeartLang | MERL | D-BETA |
|---|---|---|---|---|---|---|
| PRIE | 0.717 0.229–0.983 | 0.681 0.141–0.977 | 0.843 0.566–0.994 | 0.783 0.640–0.940 | 0.935 0.841–0.991 | 0.964 0.931–0.986 |
| | / 0.039 0.001–0.148 | / 0.032 0.001–0.130 | / 0.074 0.001–0.277 | / 0.003 0.001–0.008 | / 0.232 0.003–0.672 | / 0.014 0.007–0.028 |
| PWC | 0.772 0.622–0.896 | 0.773 0.662–0.875 | 0.902 0.803–0.967 | 0.854 0.749–0.930 | 0.927 0.869–0.974 | 0.879 0.751–0.959 |
| | / 0.020 0.007–0.041 | / 0.015 0.005–0.037 | / 0.062 0.020–0.137 | / 0.019 0.009–0.038 | / 0.099 0.032–0.228 | / 0.033 0.015–0.062 |
| QTIE | 0.656 0.576–0.740 | 0.764 0.695–0.828 | 0.931 0.899–0.960 | 0.761 0.700–0.821 | 0.940 0.914–0.964 | 0.936 0.900–0.965 |
| | / 0.043 0.018–0.094 | / 0.036 0.022–0.056 | / 0.248 0.144–0.369 | / 0.037 0.021–0.061 | / 0.225 0.133–0.325 | / 0.243 0.155–0.346 |
| RAH | 0.905 0.898–0.912 | 0.989 0.986–0.991 | 0.994 0.991–0.995 | 1.000 0.999–1.000 | 0.978 0.974–0.981 | 0.992 0.989–0.994 |
| | / 0.002 0.002–0.002 | / 0.014 0.011–0.018 | / 0.024 0.018–0.033 | / 0.317 0.125–1.000 | / 0.007 0.006–0.008 | / 0.018 0.014–0.024 |
| RBBB | 0.954 0.921–0.983 | 0.957 0.928–0.980 | 0.995 0.988–0.999 | 0.921 0.895–0.944 | 0.991 0.976–0.999 | 0.998 0.997–0.999 |
| | / 0.792 0.710–0.869 | / 0.556 0.463–0.648 | / 0.945 0.908–0.972 | / 0.407 0.316–0.496 | / 0.931 0.881–0.971 | / 0.957 0.929–0.981 |
| RVH | 0.826 0.482–0.998 | 0.824 0.519–0.999 | 0.990 0.973–0.999 | 0.920 0.766–0.999 | 0.993 0.984–0.998 | 0.969 0.916–0.999 |
| | / 0.172 0.030–0.467 | / 0.165 0.012–0.468 | / 0.260 0.066–0.632 | / 0.334 0.027–0.706 | / 0.174 0.044–0.434 | / 0.281 0.028–0.655 |
| SA | 0.744 0.718–0.769 | 0.752 0.730–0.773 | 0.927 0.914–0.939 | 0.779 0.756–0.800 | 0.830 0.811–0.849 | 0.982 0.977–0.986 |
| | / 0.238 0.203–0.277 | / 0.181 0.157–0.207 | / 0.641 0.593–0.688 | / 0.274 0.235–0.315 | / 0.400 0.351–0.448 | / 0.838 0.809–0.864 |
| SB | 0.937 0.932–0.943 | 0.976 0.973–0.980 | 0.999 0.999–1.000 | 0.987 0.985–0.990 | 0.994 0.993–0.996 | 0.999 0.999–1.000 |
| | / 0.887 0.873–0.900 | / 0.953 0.945–0.961 | / 0.999 0.999–1.000 | / 0.977 0.972–0.982 | / 0.989 0.985–0.993 | / 0.999 0.998–0.999 |
| SR | 0.754 0.740–0.767 | 0.894 0.885–0.903 | 0.990 0.988–0.993 | 0.933 0.925–0.941 | 0.968 0.963–0.972 | 0.993 0.991–0.995 |
| | / 0.426 0.406–0.446 | / 0.736 0.715–0.756 | / 0.970 0.963–0.976 | / 0.825 0.805–0.845 | / 0.904 0.890–0.916 | / 0.981 0.977–0.985 |
| ST | 0.935 0.926–0.944 | 0.969 0.964–0.973 | 0.996 0.994–0.998 | 0.985 0.981–0.987 | 0.993 0.990–0.995 | 0.996 0.993–0.998 |
| | / 0.748 0.720–0.777 | / 0.838 0.813–0.862 | / 0.986 0.981–0.990 | / 0.927 0.914–0.940 | / 0.973 0.966–0.979 | / 0.988 0.984–0.993 |
| STDD | 0.871 0.823–0.917 | 0.872 0.829–0.912 | 0.970 0.956–0.981 | 0.892 0.850–0.926 | 0.966 0.952–0.978 | 0.974 0.965–0.982 |
| | / 0.142 0.099–0.197 | / 0.131 0.091–0.181 | / 0.400 0.306–0.495 | / 0.170 0.117–0.238 | / 0.415 0.324–0.512 | / 0.409 0.320–0.515 |
| STE | 0.868 0.828–0.900 | 0.745 0.688–0.801 | 0.931 0.905–0.954 | 0.797 0.755–0.836 | 0.919 0.887–0.945 | 0.911 0.882–0.939 |
| | / 0.183 0.127–0.256 | / 0.130 0.081–0.191 | / 0.309 0.230–0.397 | / 0.080 0.052–0.118 | / 0.323 0.241–0.414 | / 0.247 0.178–0.332 |
| STTC | 0.842 0.810–0.874 | 0.839 0.808–0.868 | 0.939 0.921–0.953 | 0.861 0.835–0.885 | 0.937 0.920–0.953 | 0.942 0.924–0.956 |
| | / 0.205 0.162–0.254 | / 0.181 0.142–0.230 | / 0.406 0.341–0.477 | / 0.201 0.157–0.249 | / 0.359 0.300–0.426 | / 0.378 0.314–0.448 |
| STTU | 0.716 0.587–0.829 | 0.779 0.663–0.882 | 0.864 0.788–0.926 | 0.772 0.670–0.861 | 0.912 0.837–0.959 | 0.928 0.884–0.963 |
| | / 0.049 0.011–0.129 | / 0.081 0.024–0.183 | / 0.110 0.037–0.229 | / 0.094 0.013–0.210 | / 0.176 0.066–0.326 | / 0.134 0.057–0.258 |
| SVT | 0.967 0.946–0.983 | 0.990 0.987–0.993 | 0.992 0.982–0.998 | 0.984 0.973–0.992 | 0.996 0.993–0.998 | 0.991 0.977–0.999 |
| | / 0.579 0.490–0.667 | / 0.697 0.616–0.774 | / 0.845 0.779–0.903 | / 0.749 0.675–0.819 | / 0.863 0.805–0.916 | / 0.890 0.835–0.936 |
| TWC | 0.868 0.853–0.884 | 0.849 0.835–0.863 | 0.944 0.937–0.950 | 0.848 0.835–0.862 | 0.940 0.932–0.948 | 0.916 0.906–0.926 |
| | / 0.578 0.545–0.611 | / 0.524 0.492–0.557 | / 0.761 0.735–0.787 | / 0.529 0.499–0.558 | / 0.764 0.738–0.791 | / 0.697 0.670–0.724 |
| TWO | 0.894 0.875–0.910 | 0.854 0.825–0.880 | 0.952 0.943–0.961 | 0.836 0.808–0.862 | 0.937 0.922–0.950 | 0.958 0.950–0.966 |
| | / 0.267 0.227–0.311 | / 0.282 0.231–0.338 | / 0.464 0.405–0.523 | / 0.244 0.194–0.295 | / 0.454 0.395–0.513 | / 0.485 0.426–0.548 |
| UW | 0.730 0.570–0.867 | 0.874 0.820–0.920 | 0.929 0.843–0.984 | 0.849 0.766–0.915 | 0.909 0.858–0.955 | 0.906 0.842–0.947 |
| | / 0.009 0.003–0.020 | / 0.010 0.006–0.015 | / 0.069 0.025–0.153 | / 0.011 0.005–0.021 | / 0.089 0.010–0.273 | / 0.017 0.009–0.030 |
| VB | 0.392 0.380–0.404 | 0.923 0.916–0.929 | 0.244 0.233–0.254 | 0.322 0.311–0.334 | 0.128 0.119–0.136 | 0.992 0.989–0.994 |
| | / 0.000 0.000–0.000 | / 0.002 0.002–0.002 | / 0.000 0.000–0.000 | / 0.000 0.000–0.000 | / 0.000 0.000–0.000 | / 0.018 0.014–0.024 |
| VEB | 0.657 0.330–0.960 | 0.828 0.650–0.986 | 0.887 0.675–0.995 | 0.813 0.678–0.945 | 0.877 0.692–0.995 | 0.942 0.875–0.999 |
| | / 0.013 0.001–0.041 | / 0.017 0.003–0.038 | / 0.088 0.011–0.257 | / 0.010 0.002–0.030 | / 0.068 0.008–0.202 | / 0.327 0.022–0.801 |
| VET | 0.207 0.197–0.217 | 0.834 0.825–0.843 | 0.911 0.904–0.918 | 0.618 0.606–0.630 | 0.351 0.339–0.363 | 0.999 0.999–1.000 |
| | / 0.000 0.000–0.000 | / 0.001 0.001–0.001 | / 0.002 0.002–0.002 | / 0.000 0.000–0.000 | / 0.000 0.000–0.000 | / 0.246 0.100–0.500 |
| VFW | 0.804 0.794–0.813 | 0.249 0.238–0.260 | 0.189 0.180–0.198 | 0.312 0.301–0.323 | 0.646 0.634–0.657 | 0.423 0.411–0.434 |
| | / 0.001 0.001–0.001 | / 0.000 0.000–0.000 | / 0.000 0.000–0.000 | / 0.000 0.000–0.000 | / 0.000 0.000–0.000 | / 0.000 0.000–0.000 |
| VPB | 0.691 0.600–0.774 | 0.808 0.727–0.878 | 0.955 0.933–0.975 | 0.909 0.866–0.944 | 0.905 0.863–0.941 | 0.987 0.969–0.998 |
| | / 0.084 0.028–0.161 | / 0.066 0.035–0.120 | / 0.342 0.218–0.474 | / 0.174 0.091–0.275 | / 0.141 0.080–0.218 | / 0.700 0.567–0.825 |
| VPE | 0.964 0.924–0.998 | 0.872 0.727–1.000 | 0.884 0.754–1.000 | 0.816 0.610–1.000 | 0.969 0.944–0.992 | 0.905 0.797–1.000 |
| | / 0.044 0.004–0.133 | / 0.520 0.001–1.000 | / 0.354 0.001–1.000 | / 0.520 0.001–1.000 | / 0.015 0.006–0.038 | / 0.206 0.002–0.667 |
| WPW | 0.770 0.545–0.988 | 0.893 0.821–0.958 | 0.969 0.927–0.995 | 0.905 0.796–0.984 | 0.898 0.719–0.997 | 0.948 0.894–0.987 |
| | / 0.126 0.040–0.268 | / 0.160 0.010–0.400 | / 0.322 0.093–0.581 | / 0.157 0.029–0.367 | / 0.476 0.216–0.746 | / 0.363 0.125–0.627 |

# G. Complete Evaluation on EchoNext

*Table 29.* Performance on EchoNext sub-tasks. Each cell reports AUROC (top line) and AUPRC (bottom line), each with a 95% confidence interval.

| Task | Random | CLOCS | KED | HeartLang | MERL | D-BETA |
|------|--------|-------|-----|-----------|------|--------|
| AR | 0.698 $_{0.629-0.764}$ | 0.620 $_{0.553-0.693}$ | 0.626 $_{0.553-0.692}$ | 0.627 $_{0.567-0.687}$ | 0.696 $_{0.626-0.764}$ | 0.681 $_{0.629-0.731}$ |
|  | / 0.037 $_{0.025-0.054}$ | / 0.040 $_{0.018-0.080}$ | / 0.034 $_{0.019-0.062}$ | / 0.030 $_{0.016-0.059}$ | / 0.061 $_{0.028-0.110}$ | / 0.022 $_{0.017-0.027}$ |
| AS | 0.753 $_{0.723-0.784}$ | 0.695 $_{0.661-0.729}$ | 0.763 $_{0.735-0.792}$ | 0.710 $_{0.677-0.744}$ | 0.802 $_{0.776-0.827}$ | 0.776 $_{0.751-0.800}$ |
|  | / 0.187 $_{0.152-0.226}$ | / 0.140 $_{0.112-0.169}$ | / 0.171 $_{0.143-0.205}$ | / 0.133 $_{0.110-0.161}$ | / 0.228 $_{0.189-0.269}$ | / 0.159 $_{0.135-0.190}$ |
| LVEF $\leq 45$ | 0.856 $_{0.842-0.870}$ | 0.777 $_{0.760-0.793}$ | 0.858 $_{0.846-0.873}$ | 0.826 $_{0.810-0.840}$ | 0.878 $_{0.865-0.892}$ | 0.862 $_{0.849-0.874}$ |
|  | / 0.619 $_{0.589-0.648}$ | / 0.483 $_{0.453-0.512}$ | / 0.622 $_{0.593-0.653}$ | / 0.548 $_{0.521-0.580}$ | / 0.667 $_{0.638-0.697}$ | / 0.613 $_{0.584-0.641}$ |
| LVWT $\geq 13$ | 0.742 $_{0.726-0.758}$ | 0.670 $_{0.651-0.688}$ | 0.744 $_{0.728-0.760}$ | 0.715 $_{0.699-0.731}$ | 0.744 $_{0.728-0.760}$ | 0.718 $_{0.702-0.734}$ |
|  | / 0.416 $_{0.391-0.442}$ | / 0.342 $_{0.319-0.367}$ | / 0.417 $_{0.392-0.443}$ | / 0.361 $_{0.340-0.384}$ | / 0.413 $_{0.388-0.441}$ | / 0.354 $_{0.332-0.378}$ |
| MR | 0.789 $_{0.762-0.814}$ | 0.723 $_{0.696-0.752}$ | 0.796 $_{0.772-0.819}$ | 0.744 $_{0.717-0.773}$ | 0.804 $_{0.780-0.827}$ | 0.796 $_{0.771-0.820}$ |
|  | / 0.246 $_{0.210-0.287}$ | / 0.169 $_{0.143-0.198}$ | / 0.234 $_{0.200-0.272}$ | / 0.161 $_{0.140-0.188}$ | / 0.236 $_{0.204-0.274}$ | / 0.222 $_{0.190-0.256}$ |
| PASP $\geq 45$ | 0.741 $_{0.721-0.760}$ | 0.696 $_{0.676-0.715}$ | 0.737 $_{0.715-0.755}$ | 0.712 $_{0.693-0.730}$ | 0.752 $_{0.733-0.771}$ | 0.730 $_{0.710-0.750}$ |
|  | / 0.312 $_{0.283-0.341}$ | / 0.241 $_{0.221-0.263}$ | / 0.300 $_{0.272-0.328}$ | / 0.261 $_{0.240-0.285}$ | / 0.335 $_{0.306-0.367}$ | / 0.293 $_{0.266-0.321}$ |
| PEFF | 0.716 $_{0.656-0.773}$ | 0.629 $_{0.559-0.693}$ | 0.731 $_{0.665-0.794}$ | 0.660 $_{0.592-0.722}$ | 0.718 $_{0.648-0.779}$ | 0.709 $_{0.647-0.770}$ |
|  | / 0.029 $_{0.022-0.039}$ | / 0.022 $_{0.016-0.030}$ | / 0.039 $_{0.026-0.059}$ | / 0.031 $_{0.019-0.052}$ | / 0.040 $_{0.026-0.062}$ | / 0.036 $_{0.025-0.052}$ |
| PR | 0.816 $_{0.721-0.896}$ | 0.836 $_{0.779-0.885}$ | 0.788 $_{0.681-0.884}$ | 0.772 $_{0.664-0.878}$ | 0.868 $_{0.801-0.926}$ | 0.859 $_{0.781-0.919}$ |
|  | / 0.073 $_{0.011-0.185}$ | / 0.014 $_{0.009-0.022}$ | / 0.025 $_{0.011-0.051}$ | / 0.077 $_{0.010-0.204}$ | / 0.080 $_{0.024-0.175}$ | / 0.029 $_{0.013-0.057}$ |
| RVSD | 0.841 $_{0.821-0.862}$ | 0.793 $_{0.771-0.814}$ | 0.843 $_{0.822-0.865}$ | 0.830 $_{0.810-0.850}$ | 0.856 $_{0.836-0.877}$ | 0.850 $_{0.831-0.868}$ |
|  | / 0.368 $_{0.327-0.413}$ | / 0.274 $_{0.239-0.312}$ | / 0.399 $_{0.356-0.443}$ | / 0.299 $_{0.268-0.335}$ | / 0.427 $_{0.382-0.475}$ | / 0.369 $_{0.324-0.417}$ |
| SHD | 0.805 $_{0.793-0.816}$ | 0.723 $_{0.709-0.737}$ | 0.802 $_{0.790-0.814}$ | 0.772 $_{0.759-0.785}$ | 0.812 $_{0.800-0.824}$ | 0.801 $_{0.789-0.812}$ |
|  | / 0.772 $_{0.756-0.786}$ | / 0.686 $_{0.669-0.702}$ | / 0.762 $_{0.746-0.778}$ | / 0.722 $_{0.704-0.740}$ | / 0.781 $_{0.764-0.796}$ | / 0.762 $_{0.746-0.778}$ |
| TR-MAX $\geq 32$ | 0.732 $_{0.706-0.757}$ | 0.687 $_{0.662-0.712}$ | 0.725 $_{0.698-0.751}$ | 0.714 $_{0.688-0.742}$ | 0.744 $_{0.721-0.768}$ | 0.710 $_{0.684-0.735}$ |
|  | / 0.200 $_{0.170-0.233}$ | / 0.127 $_{0.112-0.145}$ | / 0.179 $_{0.152-0.209}$ | / 0.156 $_{0.136-0.182}$ | / 0.198 $_{0.169-0.236}$ | / 0.157 $_{0.136-0.180}$ |
| TR | 0.784 $_{0.759-0.810}$ | 0.715 $_{0.686-0.744}$ | 0.778 $_{0.749-0.804}$ | 0.753 $_{0.727-0.780}$ | 0.815 $_{0.792-0.838}$ | 0.791 $_{0.766-0.815}$ |
|  | / 0.229 $_{0.197-0.267}$ | / 0.143 $_{0.124-0.165}$ | / 0.225 $_{0.191-0.261}$ | / 0.181 $_{0.155-0.211}$ | / 0.265 $_{0.227-0.304}$ | / 0.235 $_{0.199-0.271}$ |

