# OpenReview forum: "Position: Evaluation of ECG Representations Must Be Fixed"
_ICML.cc/2026/Position_Paper_Track — ICML 2026 Position Paper Track regular_

### Official Review · Reviewer_TYZM · 2026-03-13

**Significance:** 3
**Argument Clarity:** 3
**Rating:** 5
**Confidence:** 3

**Questions:**

Q1: Regarding Sec. 5.4.2 (and similarly App. D.2): It is surprising to see that omitting low-sample classes mostly affects the performance of the Random feature method. Can the authors explain why this is the case?

**Alternative Views Section:**

Yes

**Compliance With Llm Reviewing Policy A Conservative:**

Affirmed.

**Discussion Potential:**

3

**Final Justification:**

The focus of the work is rather narrow. While a slightly broader discussion would be helpful, the work as-is is already high-quality and adequate for publication.

**Paper Summary:**

The work first observes that the field of ECG representation learning has converged to a few datasets and tasks. Furthermore, evaluating methods on them does not accurately reflect their clinical usefulness. This is supported by a short literature review and various experiments. The work calls for a different set of evaluation tasks that better measure clinical utility, and experiment designs that more reliably identify meaningful representations.

**Position:**

Yes

**Position In Title:**

Yes

**Related Work:**

2

**Strengths And Weaknesses:**

**Strengths:**
- Aligning the technological development of machine learning methods to real-world impact is an important issue.
- Contrastive pretraining on time series datasets is known to be very challenging. This work contributes to the wider discussion of that.
- The claims regarding current practices and the conclusions drawn being brittle are well substantiated by a small systematic literature review and a set of empirical method evaluations across various datasets.
- The overall writing is of high quality.

**Weaknesses:**
- The relation of the issues raised to more general machine learning on ECG data (not just representation learning) is not discussed. Do similar issues exist there? Is there something to learn for one from the other? Similarly, the relationship to general time-series representation learning is underexplored.
- Using "overlapping 95% confidence intervals" as a criterion for statistical indistinguishability is not sound. Statistical tests (e.g., ranking tests) should be used instead.
- It was not clear to me how the second paragraph in Sec. 6 presents a contradictory argument to the position held.

**Minor opportunities for improvement:**
- Fig. 1 could use some visual cleanup (e.g., neat alignment and layout).
- End of Sec. 3: There are already datasets for diagnosis and patient forecasting, as described in Sec. 5, such as those by Poterucha et al. (2025), Schlesinger et al. (2022), and Bergamaschi et al. (2025). It would be useful to more clearly note this already in Sec. 3.
- Regarding the end of Sec. 4, call for better multi-label benchmarks: Alternatively to the fourth point (excluding categories), one could also call upon the dataset collectors to better avoid extreme imbalances. Excluding categories might fragment the benchmark landscape and prevent simple 1:1 comparisons.
- Tab. 1 is placed too far ahead of its first reference and discussion.
- It is difficult to quickly draw conclusions regarding the uncertainty from Table 1 (and also the following results tables). Showing a single number (such as the interquartile range) as a measure of the dispersion of results (instead of the range boundaries) might ease interpretation.
- Tab. 15: Making the description column wider would ease readability.

**Support:**

3

---

> ### Author Rebuttal · Authors · 2026-03-31
>
> Thank you for your kind words, insightful suggestions, and specific recommendations to improve our work.
>
> > “The relation of the issues raised to more general machine learning on ECG data (not just representation learning) is not discussed…”
>
> Indeed, many of the issues we discuss are not unique to representation learning and apply more broadly to ML on ECG data, including clinically meaningful endpoint selection, severe label imbalance, and the need for task-specific reporting and uncertainty quantification. These concerns are especially visible in representation learning because methods are often compared across many downstream tasks. We will revise the discussion to make this explicit.
>
> > “Similarly, the relationship to general time-series representation learning is underexplored.”
>
> We agree. As noted in the Discussion section, our focus on 12-lead ECG is a limitation. We expect similar issues to arise in evaluations of other general time-series representations, but have not studied that literature carefully enough to make claims. We will revise the discussion to emphasize this as an important direction for future work.
>
> > “Using "overlapping 95% confidence intervals" as a criterion for statistical indistinguishability is not sound. Statistical tests should be used instead.”
>
> We agree that overlap of 95% confidence intervals is not the right test for statistical significance. The appropriate test depends on the experimental setting and methods. We thank the reviewer for pointing this out. We have performed a two-sided paired permutation test for the data in Tables 1, 2, 5, and 6, and will show the results in the revised manuscript. None of these tests changed the conclusions in our position paper.
>
> > “It was not clear to me how the second paragraph in Sec. 6 presents a contradictory argument to the position held.”
>
> We apologize for the lack of clarity. The second paragraph of Section 6 was meant to address a potential objection. We argue that ECG representations should be evaluated on a broad range of clinically important tasks and, ideally, with standardized and transparent benchmarks. However, since many such endpoints are difficult to release publicly because of privacy and institutional constraints, there is a paucity of public datasets. Some might object that private endpoint benchmarks are not reproducible and therefore should not be used. We disagree, which is why we report on 3 tasks (out of 100) using private data. We will revise Section 6 to make this clearer.
>
> > “Regarding Sec. 5.4.2 (and similarly App. D.2): It is surprising to see that omitting low-sample classes mostly affects the performance of the Random feature method. Can the authors explain why this is the case?”
>
> Because macro-AUROC weights the performance on each label equally, omitting a few classes with extremely small test support can disproportionately affect whichever method happened to perform the worst on those labels. In PTB-XL FORM, Random performs especially poorly on the labels with low-sample counts (e.g., STE_ and TAB_), so removing them increases its macro-AUROC the most (+0.076). We don’t think this is an intrinsic property of the Random encoder. For example, in Appendix D.2, CLOCS changes more than Random on PTB-XL SUB (+0.022 vs. +0.013), while on RHYTHM both Random and MERL shift materially (+0.054 and +0.026). We will clarify this in the revised paper.
>
> > “Minor opportunities for improvement.”
>
> We are especially grateful for the reviewer’s attention to these details. We will revise the manuscript accordingly by improving the presentation of Fig. 1 and the tables, clarifying dataset existence in Section 3, refining the discussion of benchmark imbalance in Section 4, and considering how to improve the presentation of uncertainty in the tables.

---

> > ### Author Rebuttal · Reviewer_TYZM · 2026-04-04
> >
> > I thank the authors for their response.
> >
> > The focus of the work is rather narrow. While I still think a slightly broader discussion (general literature on ECG data, general TS representation learning) would be helpful, the work as-is is high-quality and adequate for publication.

---

### Official Review · Reviewer_y4TJ · 2026-03-15

**Significance:** 2
**Argument Clarity:** 3
**Rating:** 3
**Confidence:** 4

**Questions:**

A general domain this paper carries out is the evaluation methodology in ECG representation learning. While the critique of current benchmarks is motivating, the proposed alternative framework would benefit from stronger justification and more concrete scenario-by-scenario implementation guidance.
In particular, evaluation metrics and benchmarking strategies may need to be more closely aligned with the objectives of different tasks and their potential clinical implications. Different clinical questions may require different evaluation metrics and operating frameworks. Furthermore, model evaluation and the evaluation of clinical implications of model performance should be closely connected but also explicitly distinguished in discussion.

**Alternative Views Section:**

Yes

**Compliance With Llm Reviewing Policy A Conservative:**

Affirmed.

**Discussion Potential:**

2

**Ethics Review Area:**

["Other Expertise"]

**Paper Summary:**

This paper argues that current benchmarking practices for ECG representation learning are flawed. Most studies evaluate models using a set of public datasets (mainly PTB‑XL, CPSC2018, and CSN) and focus primarily on arrhythmia and waveform classification tasks. The authors claim that this narrow evaluation framework does not clearly align with real-world clinical practice or provide strongly relevant insights.
The paper proposes that evaluation should expand to include additional clinically meaningful tasks such as structural heart disease detection, hemodynamic state inference, and patient‑level forecasting. It also argues that current evaluation metrics are often misleading. Many studies rely heavily on macro‑AUROC, which can hide important differences between tasks and is particularly problematic in highly imbalanced datasets.
To support their argument, the authors evaluate three representative ECG pre‑training methods (CLOCS, MERL, and D‑BETA) across several datasets and tasks. They also include a randomly initialized encoder as a baseline. The results show that no single method consistently dominates across tasks, and surprisingly, a random encoder with linear probing can perform competitively on many benchmarks.
The overall claim of the paper is that benchmarking practices in ECG representation learning should be expanded and standardized to ensure that progress in the field reflects clinically meaningful improvements.

**Position:**

Yes

**Position In Title:**

Yes

**Related Work:**

3

**Strengths And Weaknesses:**

Strengths
1.	The paper addresses an important methodological issue in 12-lead ECG model evaluations that has been overlooked. Benchmark design strongly influences how model performance is interpreted, and the critique of current ECG benchmarking practices is timely.
2.	The work strives to examine a notable theme: the mismatch between current machine learning benchmarks and clinically meaningful objectives. This perspective is valuable for medical AI research.
3.	Evaluating multiple models across several datasets and task categories (e.g., PTB-XL, CPSC2018, EchoNext) demonstrates that model rankings can vary substantially depending on the evaluation setup, metric choice, and label composition.
4.	The observation that randomly initialized encoders can perform competitively is interesting and raises important questions about whether current benchmarks are sufficiently challenging.
5.	The recommendations regarding better reporting practices (e.g., per-task metrics, uncertainty estimates, and appropriate baselines) are practical and could improve evaluation rigor.

Weaknesses
1.	The paper does not introduce a new model or algorithm. As a result, the technical novelty is limited, and the contribution is mainly methodologically advocated.
2.	Some of the evaluation tasks rely on private datasets. This reduces reproducibility.
3.	While the paper proposes a new evaluation framework, it does not clearly explain how this framework was derived. In particular, it is not clear why the specific endpoints (structural disease, hemodynamics, and patient forecasting) were selected as the key task categories.
4.	The discussion on evaluation metrics may still be incomplete from a clinical perspective. While the paper emphasizes AUROC and AUPRC, the most relevant metric may vary depending on the specific clinical task and objective. For example, in certain screening or detection scenarios, clinicians may prioritize sensitivity or other task-specific metrics, rather than relying solely on global summary metrics.
5.	The paper recommends reporting bootstrap-based 95% confidence intervals to better characterize uncertainty. While this aligns with statistical practices commonly used in clinical research, the practical utility of such intervals may be limited when datasets are extremely large, in the case of ECG data sources, since the confidence intervals may become very narrow and provide limited additional interpretability.
6.	The use of a randomly initialized encoder as a baseline is an interesting design choice, but the interpretation of this comparison is not fully discussed. For example, does it raise any indication on the complexity of the tasks? Additionally, while it highlights that improvements between models may not always be substantial, it does not fully address how statistical significance or practical significance should be interpreted when datasets are extremely large.

**Support:**

3

---

> ### Author Rebuttal · Authors · 2026-03-31
>
> Thank you for highlighting that our paper treats an important and overlooked problem, and especially for pointing out opportunities for improvement.
>
> > “The paper does not introduce a new model or algorithm…”
>
> As a position paper, our goal is to identify limitations in current ECG representation learning evaluation and advocate for improved practice, not to present new methods or techniques. We will state this clearly.
>
> > “Some of the evaluation tasks rely on private datasets…”
>
> Of the >100 unique tasks in our study, only three endpoints — mPCWP, mPA, and 1YR-HF — use private data. We included these endpoints because they are clinically important and not covered by public 12-lead ECG benchmarks. We argue that creating public datasets for such endpoints should be a priority, to ensure that robust ECG representations are designed. We will clarify this.
>
> > “While the paper proposes a new evaluation framework, it does not clearly explain how this framework was derived…”
>
> We apologize for the lack of clarity. We began with the observation that current ECG representation benchmark methods focus on arrhythmia and waveform-abnormality tasks. The ground-truth labels for these tasks are obtained from inspection of an ECG by an expert; this implies that all information needed for assigning the label is explicitly contained in the signal. However, clinicians want to use the ECG to predict complex endpoints that cannot be derived solely from the ECG via visual inspection (e.g. structural disease, hemodynamics, and patient forecasting). Our point is that these complementary endpoints, which are not functions of the ECG alone, are needed for a complete evaluation of ECG representations. The specific endpoints we mention are examples that fall into this category, and were selected from common tasks in the supervised ECG literature. These tasks are a starting point; we did not intend that all future benchmarking should be limited to (or necessarily include) them. We will revise the paper to make this explicit and clarify that the field should converge on a broader community consensus regarding which ECG-relevant tasks belong in a standard benchmark.
>
> > “The discussion on evaluation metrics may still be incomplete from a clinical perspective… clinicians may prioritize sensitivity or other task-specific metrics…”
>
> We agree that the relevant metrics are often task dependent. The advantage of AUROC and AUPRC is that they are threshold independent. Other metrics such as sensitivity, specificity, PPV, and NPV require thresholds. A more complete analysis would include such metrics, and we will discuss this in the revision. We will also clarify that evaluation should be matched to the intended use case. For example, diagnosis versus forecasting and screening versus confirmatory use require different metrics, operating points, and benchmarking strategies. We will also distinguish model performance from its clinical implications. The former concerns predictive performance, uncertainty, and comparison to baselines, while the latter depends on the deployment setting and the relative costs of errors.
>
> > “The paper recommends reporting bootstrap-based 95% confidence intervals… the practical utility of such intervals may be limited when datasets are extremely large…”
>
> We agree that when the dataset is large, 95% confidence intervals may be deceptively narrow. In ECG benchmarking, the power of overall dataset size is often mitigated by class imbalance. For example, under the standard PTB-XL protocol, 13 labels have fewer than 10 positive test examples, which can materially affect macro-level summaries and rankings. We therefore view confidence intervals as a useful default because they reveal when apparent differences may be sampling noise. However, as you suggest, the overlap of marginal 95% confidence intervals is not uniformly valid and precise statistical testing is needed to quantify differences between performance metrics. The appropriate test depends on the experimental setting and methods. We will revise the paper to make this point explicit.
>
> > “The use of a randomly initialized encoder as a baseline is an interesting design choice, but the interpretation of this comparison is not fully discussed…”
>
> We will revise our manuscript to clarify. Strong performance using a random encoder, in our view, suggests that the underlying task is simple. The random-encoder baseline helps contextualize whether gains from pretraining are meaningful in practice, e.g., when datasets are so large that uncertainty is small. Please refer to our response to reviewer RFEd, where we include additional experiments to characterize the randomly initialized encoder.

---

> > ### Author Rebuttal · Reviewer_y4TJ · 2026-04-06
> >
> > I will raise my score to be weak accept.

---

### Official Review · Reviewer_RFEd · 2026-03-16

**Significance:** 3
**Argument Clarity:** 3
**Rating:** 4
**Confidence:** 3

**Questions:**

1. The random encoder baseline is an important part of the paper’s case. How robust is this finding to alternative backbones, preprocessing pipelines, and probing protocols? Put differently, how confident should readers be that this is a general lesson rather than an artifact of the specific setup studied here?
2. The experimental section evaluates three representative methods. To what extent do the authors believe their conclusions about ranking instability and baseline competitiveness generalize to the broader set of ECG representation-learning papers surveyed in the manuscript?

**Alternative Views Section:**

Yes

**Compliance With Llm Reviewing Policy A Conservative:**

Affirmed.

**Discussion Potential:**

3

**Final Justification:**

Thanks for the authors' response, I would like to retain the positive score.

**Paper Summary:**

This paper argues that current evaluation of 12-lead ECG representation learning is inadequate and should be revised. The authors claim that existing practice is too focused on public arrhythmia and waveform tasks, relies heavily on macro-AUROC point estimates, rarely reports uncertainty, and often omits random encoder baselines.
The paper supports this position by reviewing current evaluation practice, proposing a broader set of clinically meaningful downstream tasks, and recommending improved reporting standards. Empirically, it compares three pretraining methods and a random encoder across six settings, showing that rankings can change under broader and more rigorous evaluation.

**Position:**

Yes

**Position In Title:**

Yes

**Related Work:**

4

**Strengths And Weaknesses:**

Strength：
1.The paper states a clear and actionable position and specifies what should change, namely task breadth, task-level reporting, uncertainty quantification, and use of a random baseline.
This is appropriate for a position paper and easy for readers to understand.
2.The paper usefully separates clinical relevance from benchmark convenience. The paper addresses at least two credible objections: first, that some clinically meaningful targets may be inherently noisy and unsuited to benchmarking; second, that some valuable endpoints may not be publicly releasable. This study places greater emphasis on clinical applicability.

Weakness:
1.Since the competitiveness of the random encoder is central to the paper’s argument, it would strengthen the position if the authors more deeply analyzed why this happens, when it is expected, and how sensitive it is to architecture, data preprocessing, or linear evaluation setup. The Discussion acknowledges this point, but the treatment remains brief.
2.The paper makes field-level claims about ECG evaluation practice, but the experimental section evaluates only three representative pretraining methods plus a random baseline. This is enough to demonstrate that the issue is real, but perhaps not enough to fully establish how broadly the observed ranking instability extends across the larger literature.

**Support:**

3

---

> ### Author Rebuttal · Authors · 2026-03-31
>
> Thank you for the suggested experiments and highlighting that we presented a clear and actionable position with well-substantiated empirical claims.
>
> ---
> > To address the comments beginning with “Since the competitiveness of the random encoder…” and “The random encoder baseline is an important part of the paper…”
>
> To better characterize the random baseline, we reran our evaluation using 36 additional random encoders – produced using different preprocessing pipelines and network backbones:
>
> - 3 ECG sampling rates (Hz): 100, 250 or 500
> - 2 ECG filtering methods: None, Bandpass (order-5 Butterworth bandpass filter from 0.5 to 40 Hz)
> - 3 ECG normalization methods: No normalization, Z-score using dataset statistics, Z-score using per-sample (ECG) statistics
> - 2 neural network backbones: ResNet18, ViT-Medium implementation from the MERL codebase [1, 2]
>
> Our preprocessing steps were based on common configurations in the ECG modeling literature. As described in Appendix C.1, we vary feature scaling, regularization strength, and class weighting in the linear probe, selecting the probe with lowest validation loss for each model.
>
> The table below (which we will add to the paper) shows that the performance of the random encoder in the original manuscript is not simply an artifact of a single setup. We compare its macro-AUROC to the mean macro-AUROC across the 36 encoders in the grid search. We also report the standard deviation across all encoders in the grid, as well as the average performance within the ResNet and ViT subsets.
>
> | Dataset | Original | All Grid | ResNet Grid | ViT Grid |
> |---|---:|---:|---:|---:|
> | PTB-XL SUPER | 0.86 | 0.82±0.08 | 0.87±0.01 | 0.76±0.07 |
> | PTB-XL SUB | 0.84 | 0.78±0.10 | 0.86±0.02 | 0.71±0.08 |
> | PTB-XL FORM | 0.68 | 0.67±0.06 | 0.71±0.03 | 0.63±0.05 |
> | PTB-XL RHYTHM | 0.75 | 0.74±0.08 | 0.80±0.04 | 0.68±0.07 |
> | CPSC | 0.85 | 0.80±0.09 | 0.87±0.02 | 0.72±0.08 |
> | CSN | 0.78 | 0.75±0.09 | 0.82±0.02 | 0.69±0.08 |
> | EchoNext | 0.77 | 0.75±0.03 | 0.77±0.01 | 0.73±0.03 |
>
> Unsurprisingly, the backbone affects performance, as ResNet18 outperformed ViT-Medium. However, while the exact level of the random baseline is setup-dependent, its competitiveness is not an artifact of the original configuration. Across a broad grid of reasonable choices it remains a non-trivial baseline, and often competitive with at least some pretrained methods.
>
> ---
> > To address the comments beginning with “The experimental section evaluates three representative methods…” and “The paper makes field-level claims about ECG evaluation practice…”
>
> To further assess our conclusions, we extended our evaluation to two distinct methods from our survey: HeartLang [3] and ECGFM-KED [4]. HeartLang is a unimodal self-supervised method that tokenizes ECGs into heartbeat-level units and pretrains using reconstruction and masked-token prediction. ECGFM-KED is a multimodal ECG-text model that aligns ECGs with reports and enriches the text supervision with clinical knowledge generated by a large language model.
>
> As an extension of Table 1, we report macro-AUROC with 95% CIs for each dataset:
>
> | Dataset | HeartLang | ECGFM-KED |
> |---|---|---|
> | PTB-XL SUPER | 0.82 [0.81, 0.83] | 0.88 [0.88, 0.89] |
> | PTB-XL SUB | 0.82 [0.80, 0.84] | 0.87 [0.84, 0.90] |
> | PTB-XL FORM | 0.70 [0.67, 0.73] | 0.77 [0.74, 0.79] |
> | PTB-XL RHYTHM | 0.84 [0.80, 0.87] | 0.84 [0.81, 0.86] |
> | CPSC2018 | 0.85 [0.84, 0.86] | 0.90 [0.89, 0.91] |
> | CSN | 0.78 [0.76, 0.80] | 0.85 [0.84, 0.87] |
> | EchoNext | 0.74 [0.73, 0.75] | 0.77 [0.75, 0.78] |
>
> As an extension of Table 2, we report representative task-level AUROC/AUPRC results with 95% CIs. The full set of results will be included in the revision:
>
> | Model | Norm | LVH | LVEF<=45 | mPCWP | 1YR-HF |
> |---|---|---|---|---|---|
> | HeartLang | 0.87 [0.86, 0.89] / 0.82 [0.79, 0.84] | 0.82 [0.79, 0.85] / 0.37 [0.31, 0.42] | 0.82 [0.81, 0.84] / 0.55 [0.52, 0.58] | 0.69 [0.66, 0.72] / 0.71 [0.68, 0.75] | 0.79 [0.78, 0.79] / 0.58 [0.58, 0.59] |
> | ECGFM-KED | 0.91 [0.90, 0.92] / 0.88 [0.86, 0.89] | 0.94 [0.93, 0.95] / 0.68 [0.62, 0.74] | 0.86 [0.85, 0.87] / 0.62 [0.58, 0.65] | 0.72 [0.69, 0.75] / 0.74 [0.70, 0.77] | 0.81 [0.81, 0.81] / 0.63 [0.63, 0.64] |
>
> These results support the same qualitative conclusions as the methods reported on in the original manuscript.
>
> 1. **Ranking instability persists.** Under macro-AUROC, both HeartLang and ECGFM-KED trail MERL and D-BETA on the standard benchmark summaries. However, task-level results tell a different story: ECGFM-KED achieves the best performance on LVH.
>
> 2. **Baseline competitiveness persists.** HeartLang falls below Random on several settings, including two PTB-XL splits, Norm, LVH, and mPCWP, and both new methods fall below Random on EchoNext and LVEF.
>
> [1] https://arxiv.org/abs/2010.11929
>
> [2] https://arxiv.org/abs/2403.06659
>
> [3] https://arxiv.org/abs/2502.10707
>
> [4] https://www.cell.com/cell-reports-medicine/fulltext/S2666-3791(24)00646-3

---

> > ### Author Rebuttal · Reviewer_RFEd · 2026-04-03
> >
> > Thanks for the authors' detailed response; my concerns have been well addressed.

---

### Decision · Program_Chairs · 2026-04-30

**Decision:**

Accept (regular)

**Comment:**

ECGs are the go-to first test in cardiology.  Reading and predicting from ECGs is therefore an vitally important problem, if narrow in the scope of ML/AI.  Anything we can do to press for more data (patient data is always hard to publish because of privacy concerns) and more applications to ECG data is greatly needed.  The paper makes a compelling case, points to new tasks, and points to new data sets.  Thus this is an important position to stake.

There was great discussion between reviewers and authors, and we seem to have reached consensus broadly on this one.